# Edge-focused wire arc additive manufacturing: Method development with ANN-based stress–strain and mass-efficiency

Tran Le Hong Ngoc[1], Ha Thi Xuan Chi[1], Van-Thuc Nguyen[2], Pham Son Minh[2]*

**1** School of Industrial Engineering and Management, International University—Vietnam National University HCMC, Ho Chi Minh City, Vietnam, **2** Faculty of Mechanical Engineering, HCMC University of Technology and Engineering, Ho Chi Minh City, Vietnam

* minhps@hcmute.edu.vn

## Abstract

This study evaluates Edge-Focused Wire Arc Additive Manufacturing (EF-WAAM) for CT38 steel using an ER70S-6 filler. EF-WAAM employs an edge-guided tool-path with a prescribed travel angle to localize heat input per unit length, narrow the heat-affected zone, and mitigate residual stress relative to conventional WAAM. Under standardized 3-point bending with identical specimen geometry and span, the maximum flexural stress of EF-WAAM builds ranges from 2,414.21 to 3,338.11 MPa (n = 5 per condition). The best case improves 171% over the CT38 substrate (1,231.5 MPa) and exceeds typical values for conventional WAAM (< 2,000 MPa). Mass efficiency, reported as strength-to-density ($\sigma/\rho$), reaches 420.4 MPa·cm³·g$^{-1}$, representing gains of 40.4% versus the substrate and 83.3% versus conventional WAAM. An artificial neural network (ANN) maps process variables—current, step-over distance, travel angle, travel speed, layer thickness, and strain—to stress and reconstructs full stress–strain curves with high agreement on training, validation, and held-out test sets. ANOVA/S-N and sensitivity analyses indicate layer thickness is the dominant factor within the explored window, with beneficial interactions from travel speed and current that moderate thermal gradients. The study also demonstrates the feasibility of internal features (e.g., 3D spiral channels) while maintaining controlled thermal fields. Overall, EF-WAAM delivers higher flexural strength and improved mass-specific performance within a standardized, reusable evaluation pipeline, offering a transferable workflow for other WAAM variants. Beyond a single case, we provide a reusable evaluation pipeline, actionable parameter windows, and cross-variant metrics ($\sigma/\rho$, AER) together with an ANN routine that reconstructs full stress–strain curves from process vectors. These assets enable practitioners to transfer the method to related WAAM variants without additional sensing or bespoke hardware.

**Data availability statement:** All relevant data are within the manuscript.

**Funding:** The author(s) received no specific funding for this work.

**Competing interests:** The authors have declared that no competing interests exist.

## 1. Introduction

Additive manufacturing (AM) enables the fabrication of metal components with complex geometries, high material efficiency, and reduced waste compared with conventional casting or machining [1]. Metal AM technologies are now applied to steels, aluminum, titanium, and nickel-based alloys across aerospace, automotive, medical, and energy sectors, where they shorten development cycles and allow the consolidation of multi-part assemblies into single components [2,3]. Among metal AM processes, selective laser melting (SLM) and electron beam melting (EBM) rely on powder feedstocks, while directed energy deposition (DED) uses either powder or wire. Within DED, wire arc additive manufacturing (WAAM) is particularly attractive because of its low equipment cost, high deposition rate, and suitability for large-scale structures [4]. Nevertheless, challenges such as porosity, cracking, residual stress, and thermal distortion still limit part quality and broader industrial adoption [5].

WAAM employs an electric arc to melt a continuously fed wire and deposits successive beads to build near-net-shape components layer by layer (Fig 1) [6,7]. Several WAAM variants address different materials and productivity requirements. Gas metal arc–based WAAM (GMAW-WAAM or MIG-based WAAM) offers high deposition rates and is widely used for carbon and stainless steels and large structures [8, 9]. Gas tungsten arc–based WAAM (GTAW-WAAM) provides improved arc stability and bead control, making it suitable for higher-strength alloys and tooling [10,11]. Plasma arc WAAM (PAW-WAAM) further constricts the arc and enables precise penetration control for nickel-based and high-alloy steels [12,13]. The integration of WAAM systems with robotic manipulators or CNC platforms enhances positional accuracy and enables complex toolpaths and contoured geometries [14].

Despite these developments, achieving consistent geometric fidelity and repeatable mechanical properties remains challenging [15]. Key process variables—including current, arc length or offset, travel speed, travel angle, and heat input per unit length—strongly influence bead geometry, interlayer fusion, microstructural evolution, and the extent of the heat-affected zone (HAZ). In conventional WAAM, full-surface deposition generates steep thermal gradients that promote residual stress, distortion, and heterogeneous microstructures [16,17]. Heating large substrate areas simultaneously leads to nonuniform temperature fields; during cooling, differential contraction induces intrinsic stresses that degrade bead bonding and dimensional accuracy. Surface roughness is typically high, requiring extensive post-processing that increases cost and offsets productivity advantages [18]. Additional defects such as porosity and cracking can arise from mismatched cooling rates between adjacent layers, particularly when newly deposited material interacts with earlier layers at different thermal states [19,20]. Stress localization at interlayer boundaries reduces mechanical reliability, while cumulative heat input during multilayer deposition results in nonuniform microstructure and residual stress within the build volume [3,21]. Consequently, applications requiring tight tolerances and repeatable properties—such as aerospace and automotive components—remain difficult to achieve with conventional WAAM [22,23]. Improving WAAM performance therefore requires more targeted control of heat input and material placement. Promising strategies include refined control of current, travel

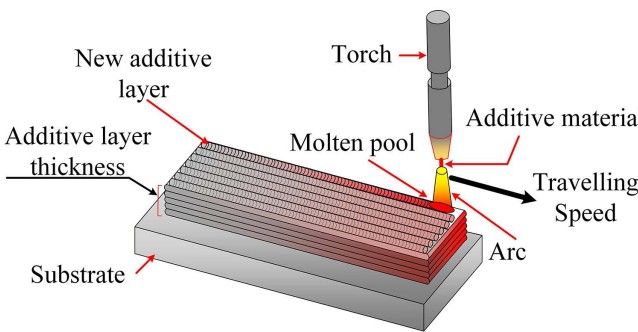

**Fig 1. Traditional WAAM.**

speed (F), travel angle, and heat input per unit length, as well as tailored consumables and in-situ monitoring to stabilize bead formation and reduce distortion [24]. These approaches aim to localize thermal effects, improve fusion quality, and enhance mechanical robustness without sacrificing productivity. To address these challenges, this study introduces Edge-Focused Wire Arc Additive Manufacturing (EF-WAAM). Unlike conventional WAAM, which deposits material across the entire surface, EF-WAAM concentrates deposition along the periphery of a pre-shaped metallic substrate. By localizing heat where it contributes most to structural performance, EF-WAAM reduces the volume subjected to repeated high-temperature cycles, limits distortion and residual stress, and enhances flexural response without increasing overall build density.

Fig 2a summarizes the EF-WAAM workflow as a deposit–machine–inspect loop. The process begins with substrate preparation and rigid fixturing to provide a stable heat sink. Edge-centric deposition follows, with the torch tracking outer contours at a prescribed travel angle. Key parameters include current (I), voltage (V), travel speed (F), wire feed rate, additive layer thickness (t), and interpass temperature. Placing material near the extreme fibers of the cross-section improves bending efficiency, while contact with the cooler substrate promotes rapid heat extraction, narrows the HAZ, and reduces fusion-related defects when heat input per unit length is maintained within a stable window. Intermediate machining restores a flat datum and prepares a uniform surface for subsequent layers, ensuring dimensional control.

Fig 2b illustrates an edge-reinforced component in which peripheral flanges carry bending loads efficiently with minimal added mass. Compared with conventional WAAM, EF-WAAM stabilizes the thermal field, improves dimensional fidelity, and delivers higher mass efficiency by placing material only where it provides structural benefit [25]. Fig 2c presents a process–structure–property map linking EF-WAAM parameters to microstructural outcomes and mechanical performance.

Flexural strength is a key design metric for components subjected to combined or cyclic loading in aerospace, automotive, and energy applications [26,27]. While many WAAM studies report only peak strength values, the full stress–strain response provides essential insight into stiffness, yielding, hardening, ductility, and failure behavior [28–33]. In this work, EF-WAAM is coupled with predictive modeling to evaluate and forecast flexural behavior. An artificial neural network (ANN) is trained to predict the complete stress–strain curve from process variables, enabling systematic parameter screening and geometry refinement with reduced experimental effort [34,35]. Therefore, EF-WAAM is developed to overcome the inherent limitations of full-surface WAAM by localizing heat and material deposition at the periphery of the component, where bending stresses are highest, enabling improved flexural performance and dimensional stability without increasing mass. Although this study focuses on straight-edge geometries, the EF-WAAM strategy is path-planning driven and can be readily extended to curved or closed-loop profiles using CNC or robotic motion control, which will be explored in future work

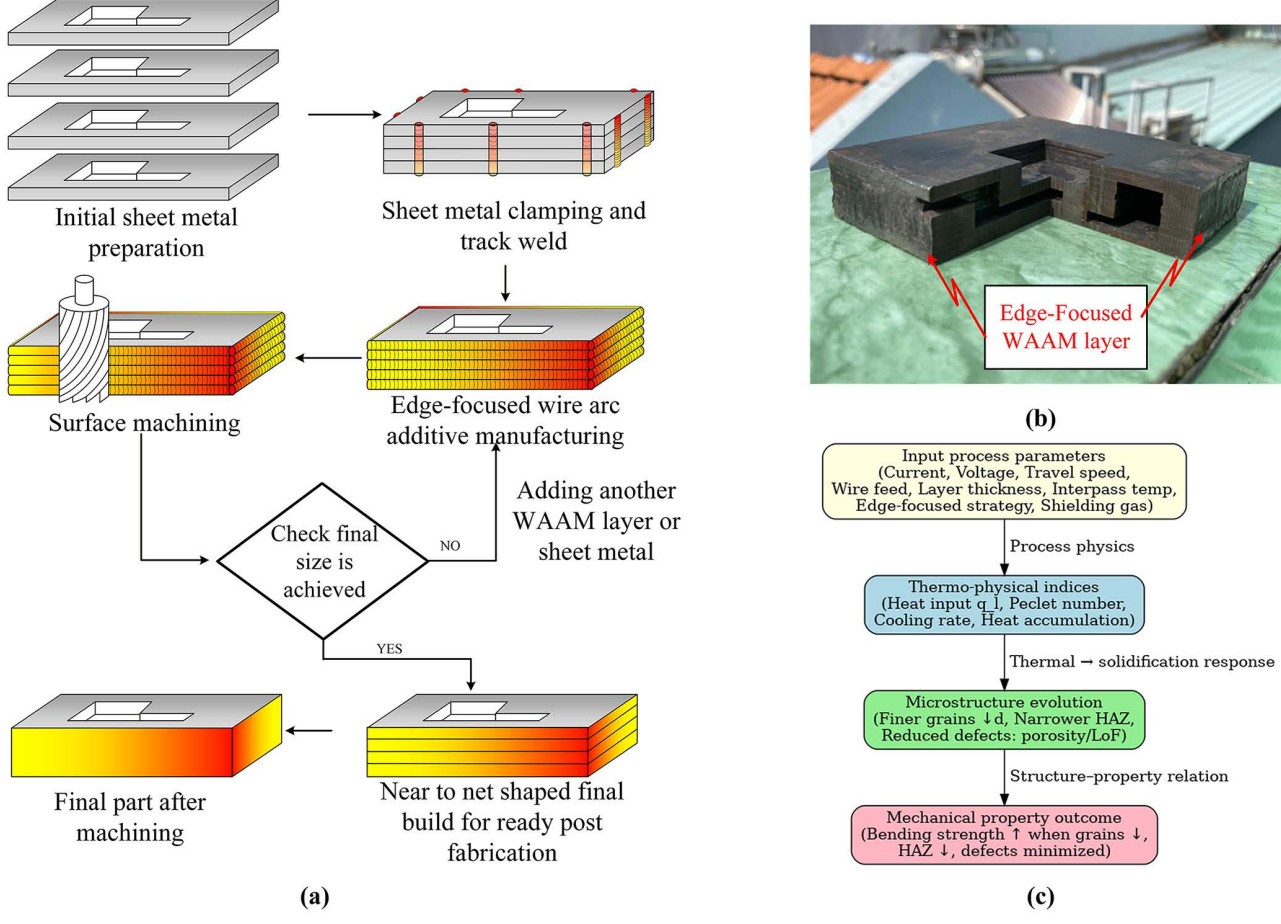

**Fig 2. Edge-Focused WAAM (a), product of Edge-Focused WAAM process (b) and the design map linking input process parameters (c).**

## 2. Research methodology

### 2.1. Materials and experimental setup

The substrate material was CT38 low-carbon steel. Its nominal composition includes C 0.17–0.25 wt%, Si 0.17–0.37 wt%, Mn 0.35–0.65 wt%, with P and S ≤ 0.045 wt%. The measured composition in this study was C 0.090 wt%, Mn 1.165 wt%, P 0.011 wt%, S 0.029 wt%, Cr 0.015 wt%, and Ni 0.012 wt%. CT38 typically exhibits tensile strength of 380–490 MPa, yield strength ≥ 235 MPa, and elongation ≥ 25%, providing good weldability and reliable bending performance. Plates were prepared with dimensions of 5 mm thickness, 7 mm width, and 270 mm length. Surface preparation included cutting or milling to size, degreasing, light grinding to remove oxides, and scribing datum lines. All surfaces were cleaned immediately prior to deposition to promote sound fusion and minimize porosity initiation.

EF-WAAM was conducted using a custom-built WAAM system comprising a three-axis CNC gantry integrated with a GMAW power source (MIG 200Y) and ER70S-6 (GM70S) solid wire (Fig 3). The CNC system executed edge-guided toolpaths with controlled travel angle and programmed travel speed F. Torch height and lateral offset were adjusted to maintain a stable arc length and consistent bead overlap. Key process parameters—including current (I), voltage (V), wire feed rate, and travel speed—were continuously logged to ensure repeatable deposition. During processing, CT38 plates

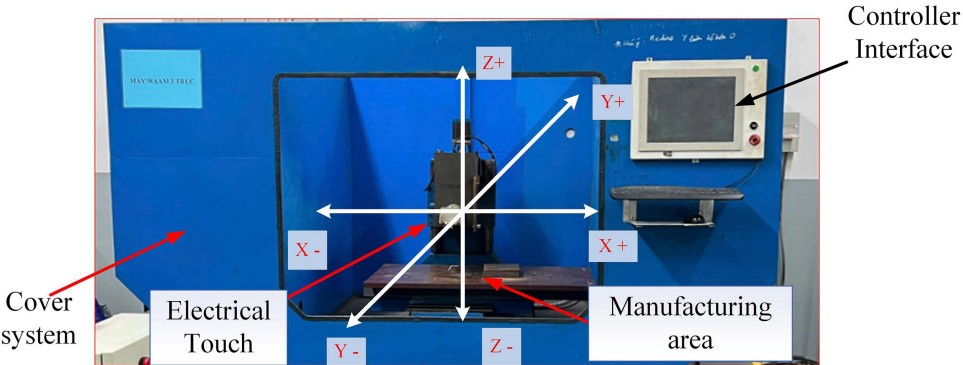

**Fig 3. Edge-Focused WAAM system.**

were rigidly clamped to a steel jig; where required, short track welds were applied to close gaps and stabilize the baseline geometry, providing a consistent heat sink and limiting distortion.

The EF-WAAM procedure followed a deposit–machine–inspect loop (Fig 4a). After substrate profiling and fixturing, reference track welds were applied along the edges, followed by calibration of torch position and process parameters. Edge-centric deposition was then performed under argon shielding (~15 L/min), with additive layer thickness maintained between 1 and 3 mm to balance fusion quality and productivity. Interim machining restored a flat datum, while dimensional checks ensured near-net accuracy. Ambient cooling between passes (5–10 min) was applied as needed to control inter-pass temperature. At the completion of multilayer edge deposition and interim finishing, the component reached the near-net geometry illustrated in Fig 4b, demonstrating that the deposit–machine–inspect loop effectively controlled dimensional growth while maintaining a uniform edge-reinforced profile suitable for subsequent specimen extraction and mechanical testing. Compared with conventional WAAM (Fig 5), EF-WAAM produced smoother edges, more uniform laminations, and reduced distortion due to localized heat input and improved thermal stability.

Specimen extraction and test geometry (Fig 6a; Fig 6b). After the EF-WAAM build reached near-net dimensions, bending specimens were extracted as shown in Fig 6a. Sections were separated by plasma cutting at 40 A and 2 mm/s to minimize heat-affected regions, followed by milling using dedicated fixtures to achieve the target geometry without inducing distortion. Fig 6b shows the specimen dimensions for three-point bending. Cover plates (20 × 9 × 250 mm) were prepared to ensure consistent stock, and final coupons were finished to 5 × 7 × 50 mm in accordance with ISO 7438:2020 and TCVN 198:2008. A single specimen geometry was used for all groups to enable direct comparison of bending strength while minimizing additional thermal input from post-processing.

**2.1.1. Experimental variables.** The selection of WAAM process parameters and their investigated ranges was based on stable processing windows widely reported for steel-based GMAW/MIG WAAM systems. The current range of 80–100 A and travelling speed of 500–900 mm/min were chosen to balance penetration depth, bead stability, and heat input per unit length, in line with previous studies showing that excessive current or low travel speed leads to grain coarsening, wider heat-affected zones, and increased residual stress, whereas overly low heat input results in lack of fusion and poor interlayer bonding [8,12,6]. The step-over distance of 2–4 mm corresponds to an overlap ratio of approximately 0.3–0.5 relative to typical bead widths of 6–8 mm reported for steel WAAM, which has been identified as an optimal range to ensure sufficient inter-bead fusion while avoiding excessive reheating and geometric instability [36,26].

In contrast to conventional WAAM studies, which generally assume a fixed torch orientation and implicitly defined layer height, the present Edge-Focused WAAM (EF-WAAM) introduces two additional process parameters: travel angle (α) and additive layer thickness (T). The travel angle (0°–20°) is a geometry-specific parameter that directly controls the

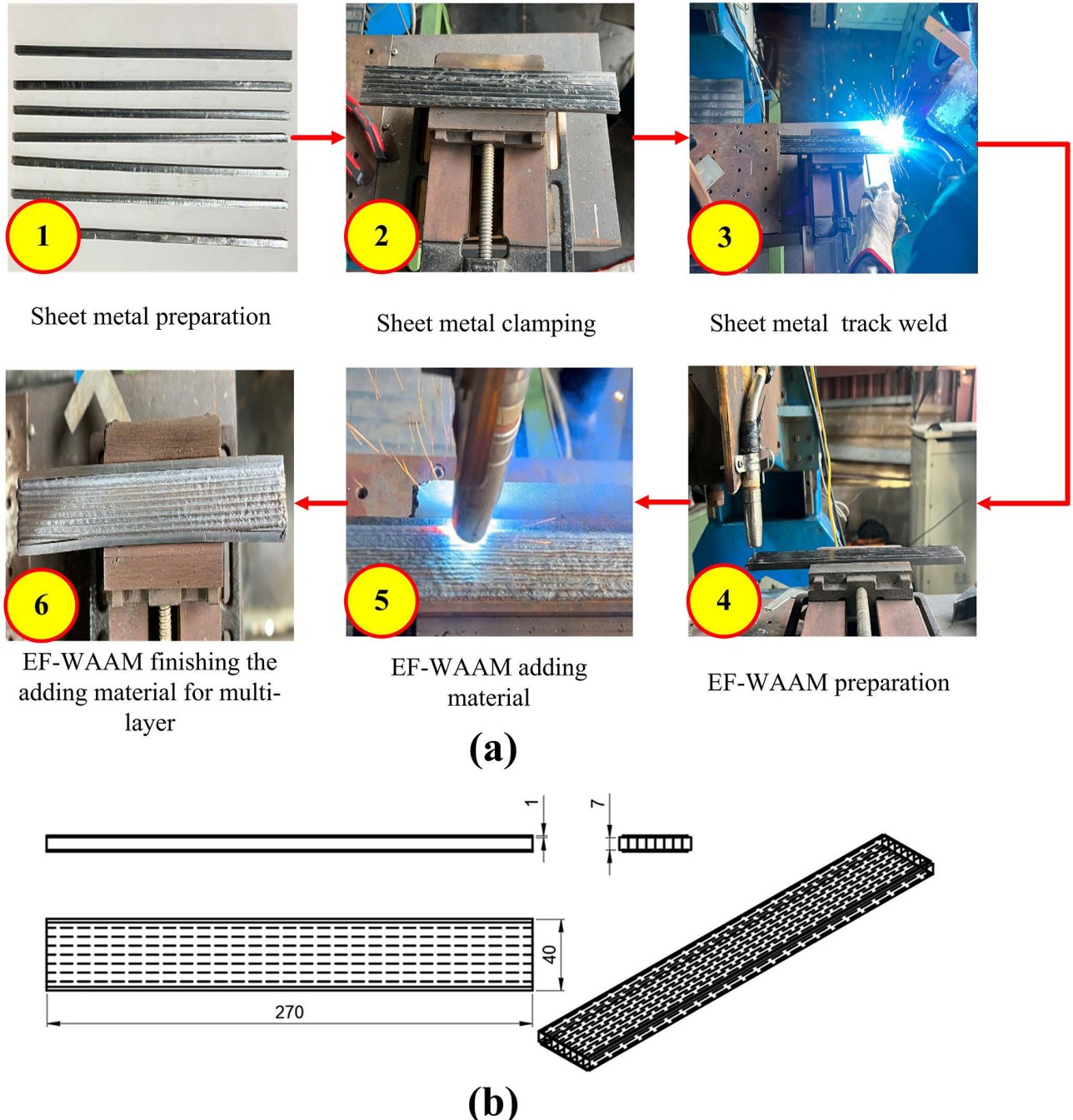

**Fig 4. Edge-Focused WAAM process (a) and part dimension at the end of EF-WAAM finishing the adding material for multi-layer (b).**

directionality of heat flow and material placement along the specimen edge, enabling localized thermal management and stress redistribution. Such an explicit angular control has not been systematically explored in prior WAAM literature, where deposition is typically normal to the substrate surface [12,37]. Similarly, the additive layer thickness (1–3 mm) was treated as an independent design variable rather than a secondary outcome of bead geometry, allowing direct control over

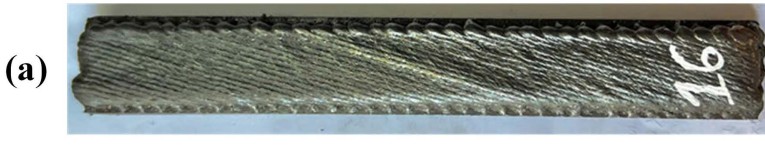

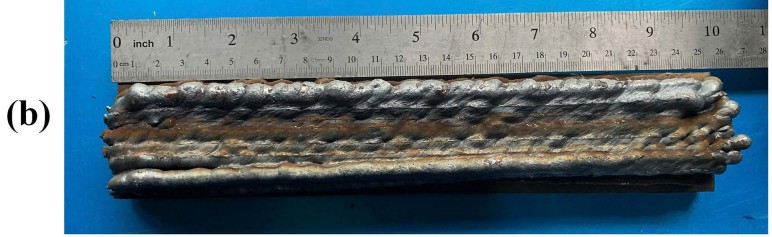

**Fig 5. Edge-Focused WAAM sample (a) vs Traditional WAAM sample (b).**

interpass heat accumulation and architectural efficiency. This range is consistent with layer heights commonly reported for stable steel WAAM builds [6,38], while enabling the investigation of thickness-driven thermal and mechanical effects specific to the EF-WAAM strategy. Overall, the selected parameter windows remain within established WAAM stability limits reported in the literature, while extending them to explore additional degrees of freedom enabled by edge-focused deposition.

The experimental variables were selected based on welding theory and empirical evidence to systematically evaluate their influence on bending strength in the EF-WAAM process. The investigated parameters and their physical meanings are illustrated in Fig 7a–d.

- Current (I, A) is a primary parameter governing heat input and melting behavior during EF-WAAM. It directly affects wire melting, bead penetration, cooling rate, and microstructural evolution. Excessive current may cause overheating and defect formation, while insufficient current can result in lack of fusion. Based on literature and preliminary trials on CT38 steel, a current range of 80–100 A (80, 85, 90, 95, and 100 A) was selected to ensure stable deposition while limiting thermal stress.

- Step-over distance (a, mm), shown in Fig 7a, refers to the spacing between adjacent deposited beads along the edge. This parameter controls bead overlap and interlayer bonding uniformity. An overly small step-over increases heat accumulation, whereas a large step-over may lead to incomplete fusion. In this study, step-over distances from 2 to 4 mm (2.0, 2.5, 3.0, 3.5, and 4.0 mm) were chosen in accordance with typical bead widths observed in steel WAAM, providing a balance between bonding quality and thermal stability.

- Travel angle (α, °), illustrated in Fig 7b, defines the orientation of the torch relative to the substrate edge and is a distinctive parameter of the EF-WAAM strategy. Unlike conventional WAAM, where the torch is typically normal to the surface, the travel angle influences material placement, heat localization, and stress distribution along the edge. A range of 0°–20° (0°, 5°, 10°, 15°, and 20°) was investigated to minimize thermal distortion while enabling effective edge reinforcement.

- Travel speed (F, mm/min), shown in Fig 7c, affects deposition rate and thermal exposure. Higher speeds reduce heat input and promote finer microstructures but may compromise bead continuity, whereas lower speeds increase heat accumulation and distortion risk. Accordingly, a range of 500–900 mm/min (500, 600, 700, 800, and 900 mm/min) was adopted to maintain stable wetting and limit weld pool temperature.

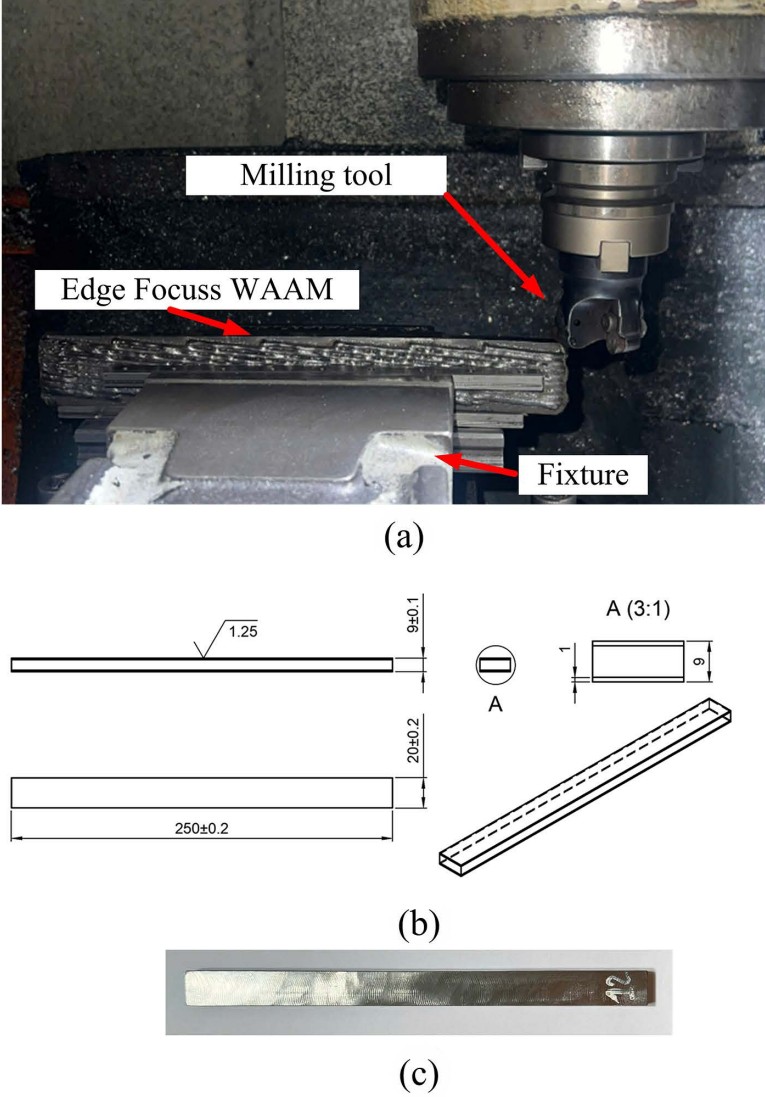

(a)

(b)

(c)

**Fig 6. Metal cutting process for bending testing sample (a) with the testing sample dimension (b) and read testing sample (c).**

- Additive layer thickness (t, mm), depicted in Fig 7d, represents the effective thickness of material added at the edge in EF-WAAM. This parameter controls cumulative heat input, build efficiency, and microstructural consistency. Layer thicknesses from 1 to 3 mm (1.0, 1.5, 2.0, 2.5, and 3.0 mm) were selected to balance productivity and thermal control, with interpass temperatures maintained within acceptable limits.

## 2.2. Theoretical background on bending strength in WAAM

Because bending induces maximum stress at the outer fibers, the three-point bending configuration used in this study inherently probes the integrity of the additive–substrate interface in EF-WAAM. Any interfacial weakness would result in early failure, which was not observed in the tested specimens. Bending strength, a critical mechanical property in WAAM, quantifies a material's ability to withstand deformation under flexural loading, which is essential for components subjected

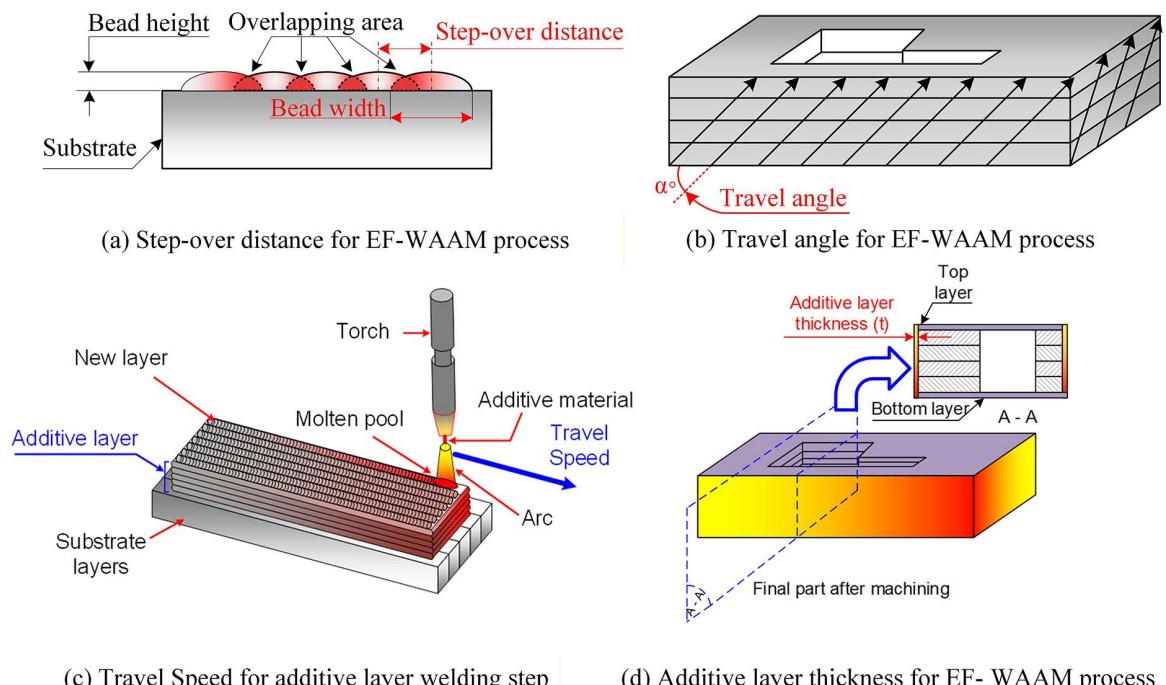

(a) Step-over distance for EF-WAAM process (b) Travel angle for EF-WAAM process

(c) Travel Speed for additive layer welding step (d) Additive layer thickness for EF- WAAM process

**Fig 7. Definition of process parameters in the EF-WAAM process.**

to complex stresses, such as aerospace frames or automotive chassis. In WAAM, bending strength is influenced by the microstructural evolution, residual stresses, and geometric accuracy resulting from the layer-by-layer deposition process. The theoretical basis for bending strength can be derived from the flexure formula for a rectangular beam under three-point bending, In this study, 'bending strength' refers to the maximum flexural stress $(\sigma_b)$ from three-point bending, computed by:

$$\sigma_b = \frac{3F_{bend}L}{2bd^2}$$

where $\sigma_b$ is the maximum bending stress (MPa), $F_{bend}$ is the applied load (N), $L$ is the span length (mm), $b$ is the width of the specimen (mm), and $d$ is the thickness (mm). This formula assumes a homogeneous material; however, in WAAM, heterogeneity due to varying thermal histories and layer interfaces complicates the stress distribution. In this research, the width and thickness of testing sample ware 9 mm and 20 mm as mention in Fig 6b.

 The main steps for preparing the bending testing samples are presented in Fig 8, ready for mechanical evaluation to assess their bending strength under controlled conditions. The specimens underwent three-point bending tests on an Instron 5982 testing machine with a maximum load capacity of 100 kN and a loading rate of 2 mm/min, as illustrated in Fig 9a. Each specimen was positioned on two support points spaced 40 mm apart, with the bending force applied at the center (Fig 9b), ensuring compliance with standard testing conditions. Stress-strain data were recorded at a frequency of 600 points per specimen, generating detailed stress-strain curves (Fig 9c).

## 2.3. Edge-focused wire arc additive manufacturing structure

Architectured materials—also known as structural hybrids of type 2 [39] —are a class of engineered materials in which geometry and spatial distribution of material phases are deliberately designed to enhance performance beyond what is

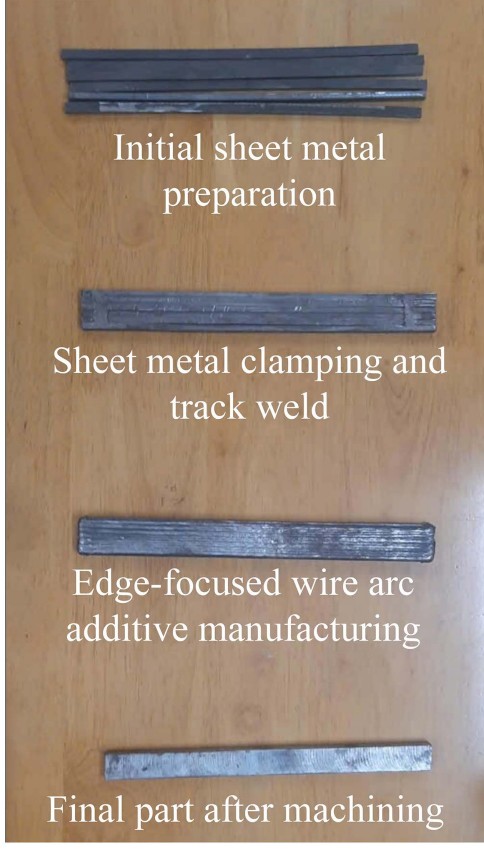

**Fig 8. Bending testing sample.**

achievable with homogeneous materials alone Fig 10. Unlike traditional composites that rely primarily on material mixing or layering, architectured materials derive their superior properties from geometry-driven optimization, particularly the distribution of strong materials in critical stress paths and the deliberate placement of lighter or voided regions elsewhere to reduce weight without compromising stiffness or strength.

One of the foundational ideas in evaluating such materials is to determine their effective density and effective mechanical performance under loading. For architectured materials with a layered or sandwich-like structure, the effective density $\widetilde{\rho}$ is given by:

$$\widetilde{\rho} = \frac{2t}{d}\rho_f + \left(1 - \frac{2t}{d}\right)\rho_c$$

Where:

- $t$ is the thickness of the face sheets (strong outer layers),

- $d$ is the total thickness of the structure,

- $\rho_f$ is the density of the face material,

- $\rho_c$ is the density of the core material (which can even be air or a lightweight filler).

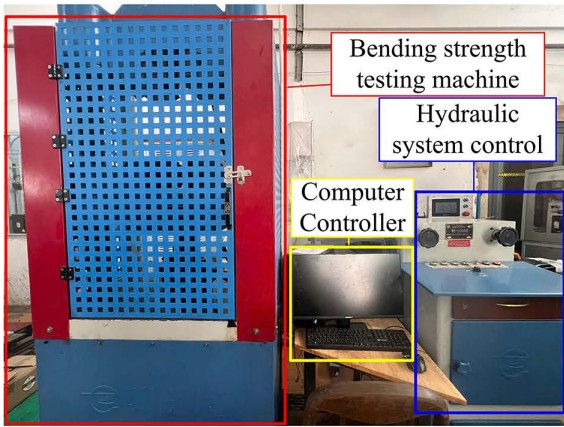

(a)

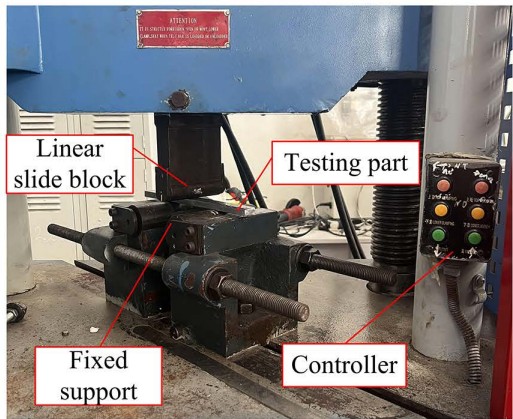

(b)

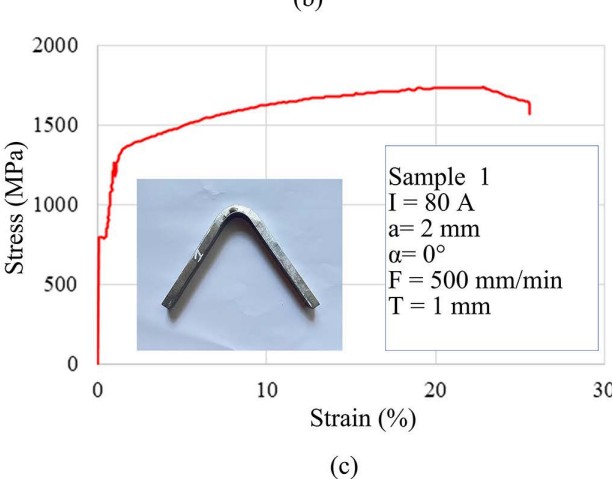

(c)

**Fig 9. Testing machine (a) and sample position for bending test (b) and stress-strain curve of bending result (c).**

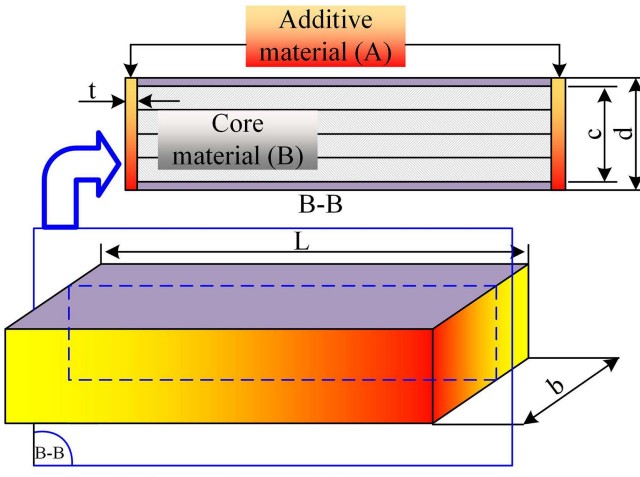

**Fig 10. Edge-Focused Wire Arc Additive Manufacturing structure.**

This formulation allows capturing how material selection and structural geometry together influence the mass efficiency of the component. In many structural applications such as beams, plates, and shells, mass efficiency—or performance per unit weight—is more critical than absolute strength.

To assess this, the material performance index $P$ is defined as:

$$P = \frac{\sigma}{\rho}$$

Where:

• $\sigma$ is the failure stress or strength of the structure,

• $\rho$ is the density.

This ratio quantifies how effectively a material carries load relative to its mass. Higher values of $P$ indicate more efficient materials or structures.

To evaluate how much more effective an architectured material is compared to a baseline (often a monolithic material), Ashby [39] introduces the Architectural Efficiency Ratio (AER):

$$AER = \frac{P_a}{P_b} = \frac{\sigma_a/\rho_a}{\sigma_b/\rho_b}$$

Where:

• Subscript $a$ refers to the architectured material,

• Subscript $b$ refers to the baseline (bulk) material.

This dimensionless metric expresses how much more (or less) efficient the designed architecture is in comparison with a conventional material. AER > 1 indicates an improved performance per unit weight.

The relative mass savings from using an architectured material are expressed using the relative weight ratio:

$$w_h = \frac{\rho_a}{\rho_b}$$

Using this, the normalized strength gain or predicted strength advantage can be approximated by:

$$\sigma_h = AER \cdot w_h \cdot \sigma_b$$

This relationship shows that the gain in strength depends not only on the architectural efficiency but also on the relative weight of the new design. In cases where significant weight is saved by strategic use of geometry, the resulting performance can far exceed that of homogeneous materials.

Finally, to assess how close the design is to the theoretical limit, Ashby proposes the material efficiency metric [39]:

$$\eta = \frac{\sigma_h}{\sigma_{max}}$$

Where $\sigma_{max}$ is the maximum achievable strength for the given material class. This metric serves as a guide to understanding how much of the material's potential is actually realized in the current design.

In summary, the design of architectured materials leverages both materials science and mechanical geometry to maximize strength-to-weight ratios, enhance stiffness, or tailor failure modes. These materials are particularly powerful in weight-sensitive applications (aerospace, automotive, biomedical devices) and are increasingly enabled by additive manufacturing, which allows precise control over geometry at multiple scales. The presented theoretical framework serves as a foundation for quantitatively comparing conventional and architectured designs, providing insight into how geometric configuration translates into mechanical advantage.

## 2.4. Experimental design using the Taguchi method

To quantify the influence of process parameters on bending strength in Edge-Focused Wire Arc Additive Manufacturing (EF-WAAM), a Taguchi L25 orthogonal array was employed. This design provides balanced coverage of multiple factor levels while maintaining a practical number of experiments, which is essential for WAAM studies where a full factorial design ($5^5 = 3,125$ combinations) is not feasible.

Five controllable process parameters were investigated within a unified workflow and identical specimen geometry. These included current (I: 80, 85, 90, 95, 100 A), step-over distance (a: 2.0–4.0 mm), travel angle (α: 0°–20°), travel speed (F: 500–900 mm/min), and additive layer thickness (t: 1.0–3.0 mm). The investigated process parameters and their corresponding levels are summarized in Table 1. The selected ranges were defined to cover stable edge deposition conditions while avoiding lack of fusion or excessive reinforcement. In the EF-WAAM context, current and travel speed primarily govern heat input and melting rate; step-over distance and travel angle control bead overlap and energy localization at the edge; and layer thickness represents a trade-off between deposition efficiency and interpass heat accumulation.

**Table 1. Researching parameters levels for EF-WAAM.**

| No. | Parameters | Level 1 | Level 2 | Level 3 | Level 4 | Level 5 |
|---|---|---|---|---|---|---|
| 1 | Current I (A) | 80 | 85 | 90 | 95 | 100 |
| 2 | Step-over distance (a (mm)) | 2 | 2.5 | 3 | 3.5 | 4 |
| 3 | Travel angle (α (°)) | 0 | 5 | 10 | 15 | 20 |
| 4 | Travelling speed (F (mm/min)) | 500 | 600 | 700 | 800 | 900 |
| 5 | Additive layer thickness T (mm) | 1 | 1.5 | 2 | 2.5 | 3 |

All 25 experimental trials were conducted in randomized order to minimize systematic drift. Each trial used the same substrate material, fixturing method, shielding conditions, and deposit–machine–inspect loop to ensure consistency. Process parameters, including current, voltage, wire feed rate, and travel speed, were logged for traceability. The response variable was bending strength obtained from three-point bending tests, with identical specimen geometry applied across all trials to enable direct comparison. Each parameter combination was repeated five times, and the mean value was used for analysis.

Following standard Taguchi practice, the signal-to-noise (S/N) ratio with a "larger-is-better" criterion was calculated to evaluate robustness and identify optimal parameter levels. Analysis included the determination of level means, main-effect trends, and a ranked recommendation of parameter settings that maximize bending strength within the observed stable processing window. In addition, analysis of variance (ANOVA) was applied to quantify the relative contribution of each parameter to bending strength variation.

Overall, the L25 design provides an efficient and statistically sound framework to assess the effects of key EF-WAAM parameters. It enables rapid identification of influential factors, supports robust process optimization, and generates a compact dataset suitable for subsequent artificial neural network modeling and cross-variant comparison.

## 2.5. Development of an ANN model for predicting stress-strain curves and bending strength

To predict the stress-strain curve and bending strength for the EF-WAAM method, this study develops an Artificial Neural Network (ANN) model, a sophisticated tool that surpasses traditional approaches limited to predicting peak stress values. The model is constructed within the MATLAB environment, leveraging its capability to model complex nonlinear relationships between input parameters and material mechanical properties. The ANN architecture follows a feedforward neural network design with three primary layers: an input layer, two hidden layers, and an output layer. Specifically, the input layer comprises six neurons corresponding to the input variables: current intensity ($I$), Step-over distance ($a$), Travel angle angle ($\alpha$), travel speed ($F$), layer thickness ($T$), and strain ($\varepsilon$). The output layer contains a single neuron representing stress ($\sigma$) in MPa, from which the maximum bending strength is inferred. The two hidden layers, each consisting of 15 neurons, are optimized to capture nonlinear relationships, employing the sigmoid activation function ($\sigma(x) = \frac{1}{1+e^{-x}}$) in the first hidden layer and a linear activation function in the second hidden layer to ensure flexibility and high accuracy in reconstructing the stress-strain curve.

Training data were collected from 25 experimental samples based on the Taguchi L25 array, yielding a total of 8692 stress-strain data points obtained from three-point bending tests conducted according to the ISO 7438:2020 standard. The dataset is partitioned as follows: 70% for training, 20% for testing, and 10% for validation, after being normalized to the range [0, 1] using the formula:

$$x_{\text{norm}} = \frac{x - x_{\text{min}}}{x_{\text{max}} - x_{\text{min}}}$$

where ($x_{\text{norm}}$) is the normalized value, ($x$) is the original value, and ($x_{\text{min}}$) and ($x_{\text{max}}$) are the minimum and maximum values of each variable, respectively. The training process employs the backpropagation algorithm with Adam optimization, defined by the weight update equations:

$$m_t = \beta_1 m_{t-1} + (1 - \beta_1) g_t$$

$$v_t = \beta_2 v_{t-1} + (1 - \beta_2) g_t^2$$

$$\theta_{t+1} = \theta_t - \eta \frac{m_t}{\sqrt{v_t} + \epsilon}$$

where $(m_t)$ and $(v_t)$ are the momentum and variance estimates, respectively, $(g_t)$ is the gradient at time step $(t)$, $(\beta_1)$ and $(\beta_2)$ are decay rates (typically 0.9 and 0.999), $(\eta)$ is the learning rate (set to 0.001), and $(\epsilon)$ is a small constant (typically $(10^{-8})$) to prevent division by zero. The mean squared error (MSE) loss function is used to evaluate the error:

$$MSE = \frac{1}{n} \sum_{i=1}^{n} (y_i - \hat{y}_i)^2$$

where $(y_i)$ is the actual value, $(\hat{y}_i)$ is the predicted value, and $(n)$ is the number of data points. Training is conducted over 1000 epochs, taking approximately 15 minutes on an Intel Core i7 CPU with 16 GB RAM, achieving a correlation coefficient ($R > 0.95$), calculated as:

$$R = \frac{\sum (y_i - \bar{y}) \left(\hat{y}_i - \bar{\hat{y}}\right)}{\sqrt{\sum (y_i - \bar{y})^2 \sum \left(\hat{y}_i - \bar{\hat{y}}\right)^2}}$$

where $(\bar{y})$ and $(\bar{\hat{y}})$ are the means of the actual and predicted data, respectively. Correlation analysis confirms a statistically significant relationship between the input parameters and bending strength, with a p-value < 0.05, ensuring the model's reliability.

Fig 11 outlines the data analysis workflow used to quantify the influence of EF-WAAM parameters on flexural behavior and to generalize the results. The workflow starts with process inputs and data capture, including current,

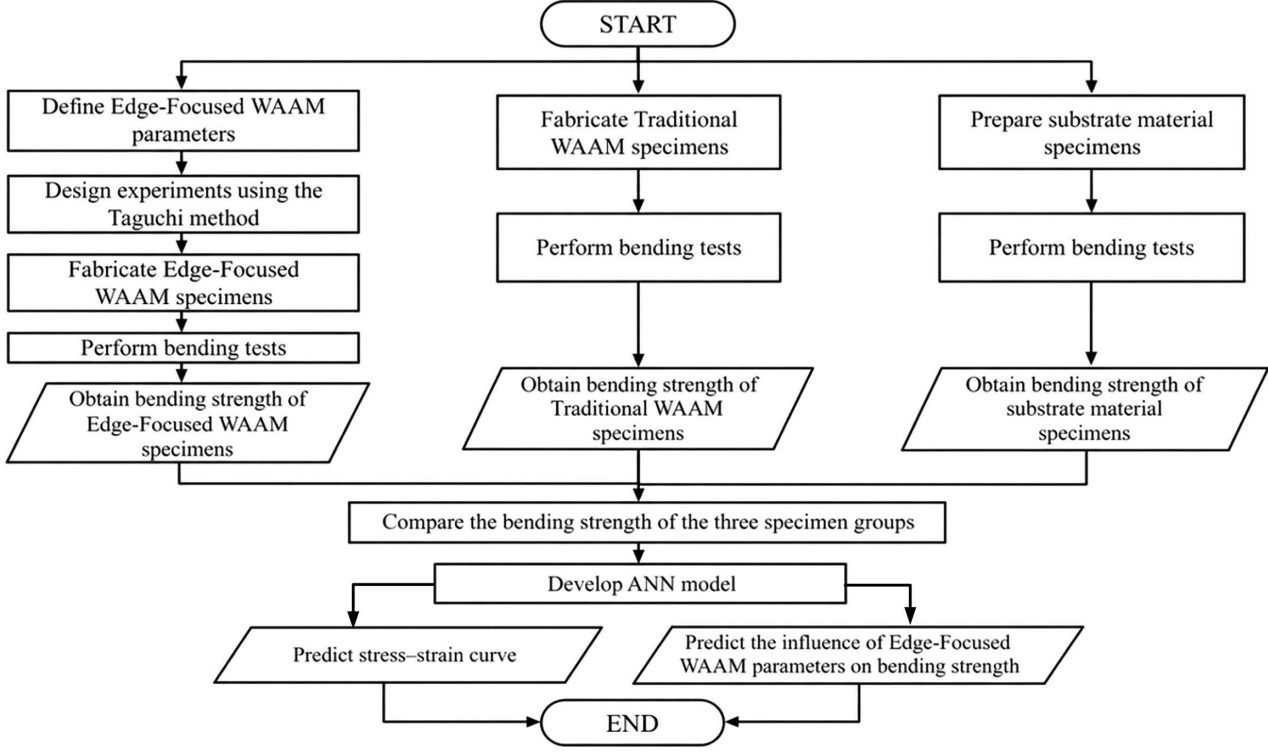

**Fig 11. Data analysis flow chart.**

voltage, travel angle, travel speed, layer thickness, wire feed rate, and interpass temperature, which together govern heat input, cooling rate, and bead fusion at the edge. Experiments are designed using a Taguchi L25 orthogonal array to efficiently span stable processing conditions while limiting the number of builds. All trials follow the same EF-WAAM deposit–machine–inspect loop and use identical specimen geometry. Flexural performance is evaluated by three-point bending and benchmarked against conventional WAAM and the substrate. Results are summarized as level means and signal-to-noise ratios, and ANOVA is applied to identify dominant parameters and their relative contributions. In the final stage, an artificial neural network (ANN) is trained to map the parameter set to the full stress–strain response, enabling prediction of bending behavior and supporting parameter optimization within the validated processing window.

## 3. Results and discussion

### 3.1. Bending strength of edge-focused WAAM

#### 3.1.1. Bending strength experimental.
This section reports flexural performance for Edge-Focused Wire Arc Additive Manufacturing (EF-WAAM) on CT38 steel using ER70S-6 filler and compares it with conventional WAAM and the substrate baseline. All coupons were tested in three-point bending with one identical specimen geometry ($5 \times 7 \times 50\,mm$) and a single protocol (ISO 7438:2020; TCVN 198:2008). Because three-point bending produces a nonuniform stress field with peak stress at the extreme fibers, the reported bending strengths (maximum flexural stresses at mid-span) can exceed the uniaxial tensile strength of the same alloy; all groups share the same geometry and span to ensure like-for-like comparison. Across the 25 Taguchi trials, Fig 12a shows EF-WAAM bending strength from 2,414.21 to 3,338.11 MPa. Several cases exceed typical values reported for conventional WAAM (<2,000 MPa). Notable outcomes include trial No. 7 at 3,338.11 MPa (peak), No. 25 at 3,169.36 MPa, and No. 1 at 2,777.15 MPa; the lower end is illustrated by No. 15 at 2,414.21 MPa. The plot includes an average line and a CT38 baseline at 1,231.5 MPa for the as-received substrate. The observed spread indicates that parameter selection—current (I), step-over distance (a), travel angle (α), travel speed (F), and layer thickness (t)—shifts flexural response within the EF-WAAM window. Fig 12b summarizes the best EF-WAAM result (3,338.1 MPa) against the CT38 substrate (1,231 MPa) and conventional WAAM (1,868.7 MPa). Under the shared geometry and method, the EF-WAAM peak is roughly 171% above the substrate reference and substantially higher than conventional WAAM. The advantage aligns with the Edge-Focused strategy: material is placed near the extreme fibers to improve section efficiency in bending, while localized heat input per unit length at the edge limits global heating and distortion. Although only three parameter combinations exceeded 3000 MPa, the distribution in Fig 12a indicates a consistent improvement over conventional WAAM across most cases

Table 2 presents the L25 layout (25 trials × 5 columns), where level assignments are distributed to keep factors orthogonal and combinations statistically balanced. All trials used the same substrate geometry and bending test protocol for apples-to-apples comparison. The 25 trials were executed in randomized order. Each trial was repeated five times; the mean of the replicates was used for analysis (and, where shown, reported as mean ± SD with the number of replicates n is 5). This design preserves clean estimation of main effects while keeping the total build count practical for EF-WAAM process evaluation.

The ANOVA results (Table 3) indicate that among the investigated process parameters, layer thickness has the most significant influence on flexural strength, contributing approximately 89.17% of the total variation with a highly significant effect ($p = 0.001$). In contrast, the effects of current, step-over distance, travel angle, and travelling speed are statistically insignificant ($p > 0.05$), each contributing less than 5% to the variation. These findings confirm that layer thickness is the dominant factor governing flexural performance in the EF-WAAM process.

Process–structure observations support this interpretation. EF-WAAM concentrates deposition along the periphery using a prescribed α\alphaα and controlled F, reducing the mass exposed to repeated high-temperature cycles. Heat is removed efficiently through the cooler bulk substrate, promoting a narrower HAZ and fewer fusion-related defects when

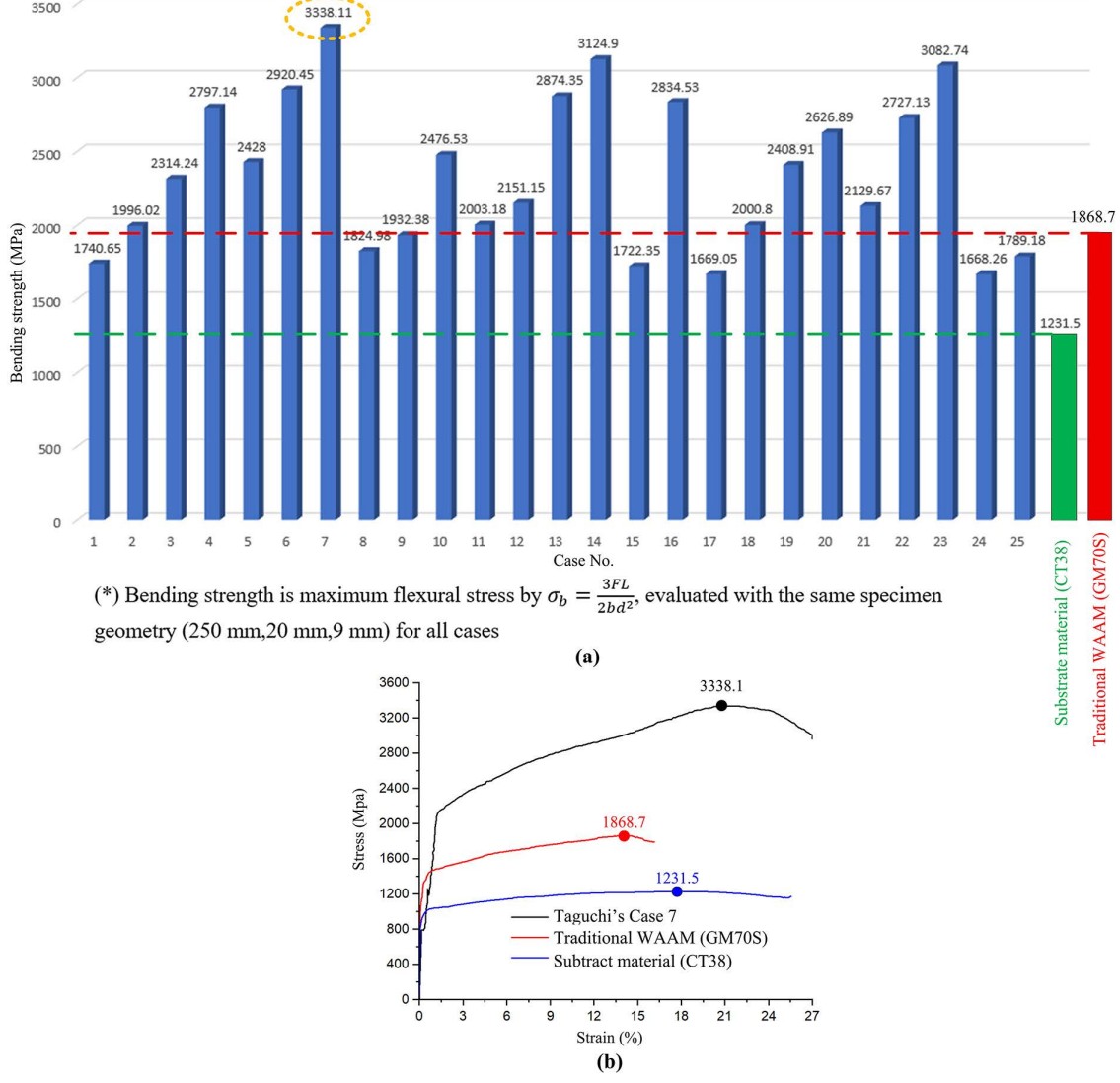

**Fig 12. Comparison between substrate, conventional WAAM, average EF-WAAM performance, and best-performing EF-WAAM case.**

operated within the stable fusion window. Microstructural evidence (Fig 13, 14) shows improved bead fusion and refined features relative to broad-area deposition, consistent with the higher bending strengths in Fig 12a. The peak value of 3,338.11 MPa is high relative to typical tensile values for carbon steel (e.g., AISI 1038/CT38) but is consistent with a flexural metric measured under one fixed geometry and span. It should be read as the maximum flexural stress in the standard three-point configuration, not as a surrogate for uniaxial tensile strength. Using one geometry and protocol for EF-WAAM, conventional WAAM, and substrate attributes the improvement to deposition strategy and parameter selection rather than specimen bias.

In summary, EF-WAAM produces bending strength well above the CT38 baseline and conventional WAAM under a common test setup. Gains are linked to Edge-Focused placement, controlled heat input per unit length, and improved bead fusion at the periphery. The range observed in Fig 12a highlights the role of parameter choice within the Taguchi

**Table 2. Experimental design for EF-WAAM and flexural strength results.**

| No. | Current I (A) | Step-over distance a (mm) | Travel angle α (°) | Travelling speed F (mm/min) | Additive Layer thickness t (mm) | Flexural Strength (MPa) |
|---|---|---|---|---|---|---|
| 1 | 80 | 2 | 0 | 500 | 1 | 1740.65±60.50 |
| 2 | 80 | 2.5 | 5 | 600 | 1.5 | 1996.02±45.60 |
| 3 | 80 | 3 | 10 | 700 | 2 | 2314.24±55.30 |
| 4 | 80 | 3.5 | 15 | 800 | 2.5 | 2797.14±60.80 |
| 5 | 80 | 4 | 20 | 900 | 3 | 2428.00±54.30 |
| 6 | 85 | 2 | 5 | 700 | 2.5 | 2920.45±59.60 |
| 7 | 85 | 2.5 | 10 | 800 | 3 | 3338.11±72.85 |
| 8 | 85 | 3 | 15 | 900 | 1 | 1824.98±49.80 |
| 9 | 85 | 3.5 | 20 | 500 | 1.5 | 1932.38±50.21 |
| 10 | 85 | 4 | 0 | 600 | 2 | 2476.53±52.66 |
| 11 | 90 | 2 | 10 | 900 | 1.5 | 2003.18±55.63 |
| 12 | 90 | 2.5 | 15 | 500 | 2 | 2151.15±60.71 |
| 13 | 90 | 3 | 20 | 600 | 2.5 | 2874.35±69.16 |
| 14 | 90 | 3.5 | 0 | 700 | 3 | 3124.90±66.55 |
| 15 | 90 | 4 | 5 | 800 | 1 | 1722.35±51.31 |
| 16 | 95 | 2 | 15 | 600 | 3 | 2834.53±50.88 |
| 17 | 95 | 2.5 | 20 | 700 | 1 | 1669.05±42.90 |
| 18 | 95 | 3 | 0 | 800 | 1.5 | 2000.80±44.58 |
| 19 | 95 | 3.5 | 5 | 900 | 2 | 2408.91±50.68 |
| 20 | 95 | 4 | 10 | 500 | 2.5 | 2626.89±60.21 |
| 21 | 100 | 2 | 20 | 800 | 2 | 2129.67±68.46 |
| 22 | 100 | 2.5 | 0 | 900 | 2.5 | 2727.13±52.84 |
| 23 | 100 | 3 | 5 | 500 | 3 | 3082.74±43.53 |
| 24 | 100 | 3.5 | 10 | 600 | 1 | 1668.26±56.31 |
| 25 | 100 | 4 | 15 | 700 | 1.5 | 1789.18±52.10 |
| Strength of substrate material | | | | | | 1231.50±30.50 |

**Table 3. ANOVA results for flexural strength (MPa) of EF-WAAM specimens.**

| Factor | F-value | P-value | % Contribution |
|---|---|---|---|
| Current (A) | 1.54 | 0.344 | 3.03% |
| Step-over distance (mm) | 1.09 | 0.466 | 2.16% |
| Travel angle (°) | 1.48 | 0.357 | 2.92% |
| Speed (mm/min) | 0.39 | 0.81 | 0.76% |
| Layer thickness (mm) | 45.26 | 0.001 ** | 89.17% |
| (*) Main effects plot for S/N ratios of flexural strength of EF-WAAM specimens are shown in Appendix, Fig B | | | |

design space. Subsequent sections relate these outcomes to factor effects and to the predictive ANN model used to screen settings without additional builds.

Table 4 compares material efficiency for three configurations: the CT38 steel substrate, conventional wire arc additive manufacturing (WAAM), and Edge-Focused WAAM (EF-WAAM) at its best condition (Case 7). The comparison uses peak

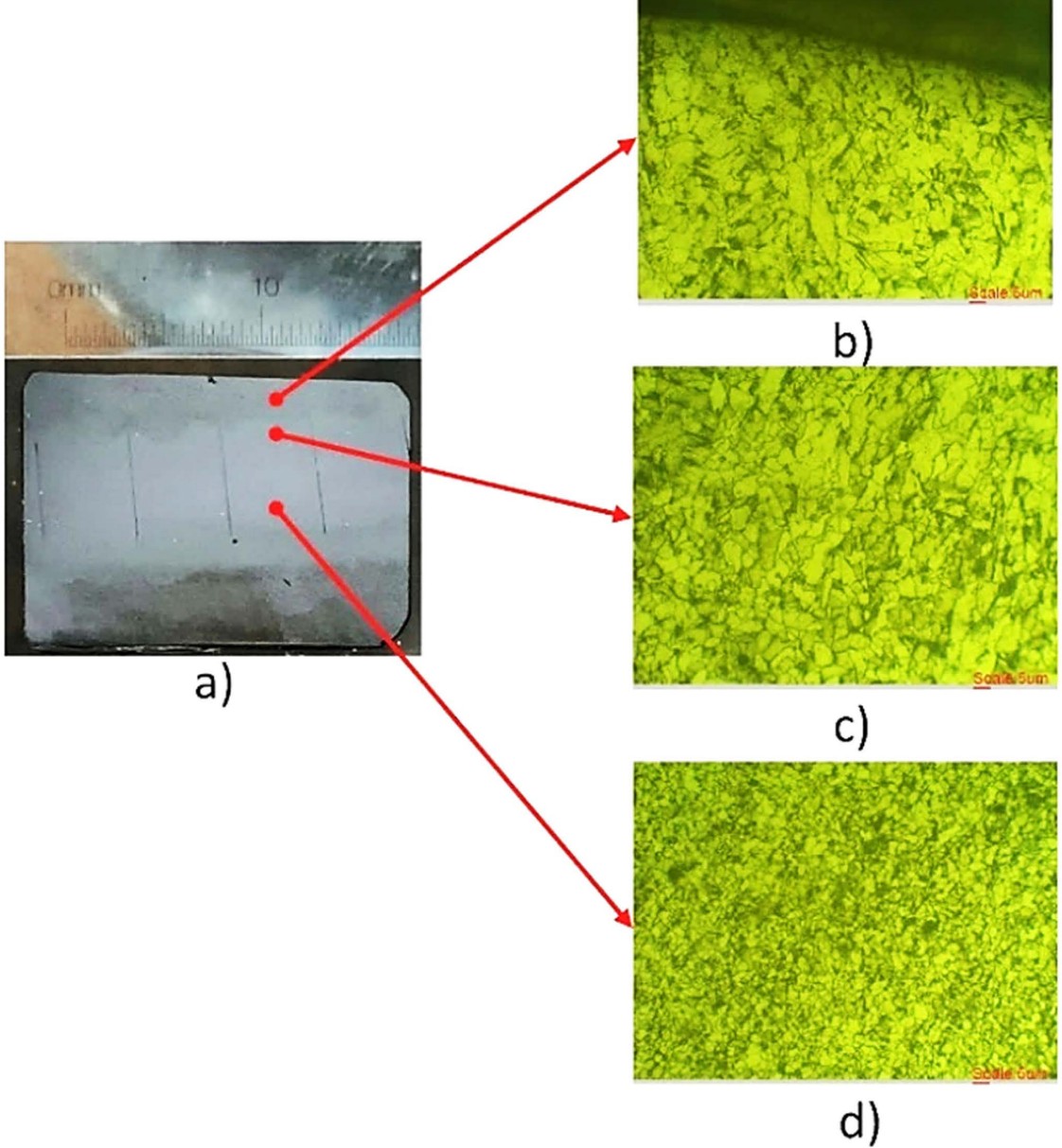

**Fig 13. Macro and Microstructure of the EF – WAAM samples.**

flexural stress (σ, MPa) under three-point bending, density (ρ, g/cm³), and a strength-to-density index (P, MPa·cm³·g⁻¹) that represents strength per unit mass. For the substrate, peak flexural stress (σ) is 1,231.5 MPa and density (ρ) is 7.85 g/cm³, which gives a strength-to-density index (P) of 299.4 MPa·cm³·g⁻¹ and serves as the baseline. Conventional WAAM shows peak flexural stress (σ) of 1,868.0 MPa at the same density (ρ), yielding a strength-to-density index (P) of 229.3 MPa·cm³·g⁻¹, lower than the substrate. The decrease in the strength-to-density index (P) aligns with broad-area deposition and higher heat input per unit length, which enlarge the heat-affected zone (HAZ) and increase microstructural variability, thereby reducing flexural response at constant mass.

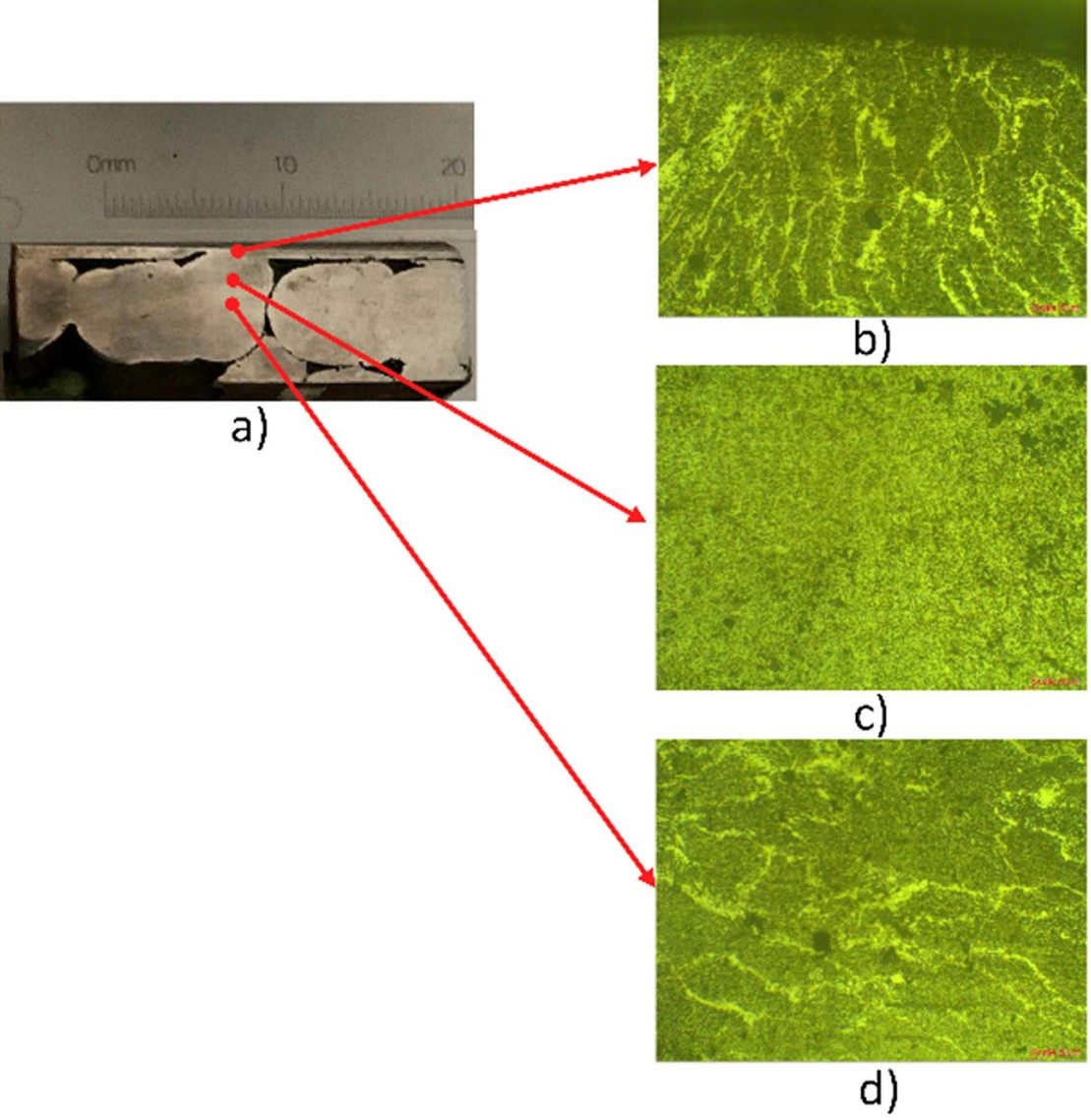

**Fig 14. Macro and Microstructure of the traditional WAAM samples.**

**Table 4. Material handling performance.**

| Case | σ (MPa) | ρ (g/cm³) | P (MPa·cm³/g) |
|---|---|---|---|
| Substrate layer (CT38) | 1231±30.5 | 7.85 | 299.4 |
| Traditional WAAM | 1868±42.1 | 7.85 | 229.3 |
| EF- WAAM (Case 7) | 3338±65.1 | 7.85 | 420.4 |

Edge-Focused WAAM localizes heat along an edge-guided path with a prescribed travel angle ($\alpha$, degrees), which limits lateral heat spread and narrows the heat-affected zone (HAZ). Under identical specimen geometry and span, Case 7 attains peak flexural stress ($\sigma$) of 3,338.0 MPa at the same density ($\rho$), giving a strength-to-density index (P) of 420.4 MPa·cm³·g⁻¹. Relative to the substrate, the strength-to-density index (P) increases by 40.4%; relative to conventional WAAM, the strength-to-density index (P) increases by 83.3%. Because density ($\rho$) is constant across groups, the higher strength-to-density index (P) for Edge-Focused WAAM reflects the gain in peak flexural stress ($\sigma$) delivered by the edge-focused toolpath and associated thermal management rather than a change in material density.

The mechanism behind the improvement is consistent with the EF-WAAM strategy. Deposition is concentrated along the periphery of the pre-shaped substrate, material is placed near the extreme fibers that carry bending, and heat input per unit length is localized at the edge. The cooler bulk substrate acts as a heat sink, supporting faster heat removal, a narrower HAZ, and fewer fusion-related defects than broad-area deposition. Under the shared specimen geometry and test method, these process–structure effects translate into higher bending strength at the same density and, therefore, higher mass-specific performance. Table 3 thus indicates that EF-WAAM is favorable for components that require high flexural strength while maintaining low mass.

### 3.1.2. Compare Novel WAAM with CT38 (AER, $w_h$, $\sigma_h$, $\eta$). AER:

$$AER = \frac{420.4}{299.4} = 1.4$$

Weight ratio:

$$w_h = \frac{7.85}{7.85} = 1.0$$

Stress prediction:

$$\sigma_h = 1.4 \cdot 1.0 \cdot 2350 = 3290 MPa \approx experiment: \ 3300 \ MPa$$

Material utilization efficiency (*assumption* $\sigma_{max}$ = 3500 MPa ):

$$\eta = \frac{3300}{3500} = 94.3\%$$

Although the Edge-Focused WAAM strategy and conventional WAAM use the same ER70S-6 filler, the Edge-Focused approach attains higher bending strength than both conventional WAAM and the CT38 substrate. The difference arises from material placement within the section rather than composition. In the Edge-Focused build, added metal is concentrated along the periphery of the transverse cross-section, where bending stresses are highest. Classical beam theory indicates that stress increases from the neutral axis toward the extreme fibers; positioning sound material at the outer surfaces increases the section modulus and moment capacity without additional mass, improving stiffness-to-mass and strength-to-mass performance at the same density.

Conventional WAAM distributes material across the entire surface. A substantial portion of mass accumulates near the neutral axis, contributing little to bending resistance while increasing inertia and total mass. At equal density, this geometry provides lower moment capacity per unit mass because much of the volume lies far from the stress peaks. The Edge-Focused layout aligns with Ashby's architected-materials perspective [39]: the section behaves as a sandwich beam, with an ER70S-6 face sheet at the extreme fibers and a CT38 core providing support. This is a direct expression of shape efficiency—using form and placement to carry load effectively.

The material-efficiency metrics support this interpretation. For the CT38 substrate, peak bending strength σ was 1,231.5 MPa and density ρ was 7.85 g/cm³; the corresponding performance index P (strength per unit mass) was 299.4 MPa·cm³/g, used here as the baseline. Conventional WAAM reached σ of 1,868.0 MPa at the same density, yielding P of 229.3 MPa·cm³/g, lower than the substrate and consistent with broad-area heating, a wider HAZ, and more variable microstructure. The Edge-Focused WAAM (Case 7) achieved σ of 3,338.0 MPa with unchanged density and P of 420.4 MPa·cm³/g. Relative to the substrate, P increased by 40.4%; relative to conventional WAAM, P increased by 83.3%. The architectural efficiency ratio (AER) was about 1.4, indicating roughly 40% higher bending load capacity at constant mass. Wire utilization approached 94%, close to the practical limit for ER70S-6 under the tested conditions, indicating that most deposited material contributed to useful section.

In summary, the performance gain of the Edge-Focused WAAM process arises from geometric intent and thermal control: placing material at the extreme fibers, maintaining heat input per unit length locally at the edge so less volume experiences repeated high-temperature cycles, enabling faster heat extraction through the cooler substrate, a narrower HAZ, and fewer fusion-related defects, and translating higher σ at fixed ρ into higher P—that is, superior mass-specific performance. Consistent with Ashby's framework [39], this illustrates shape efficiency in bending: form drives function in metal AM when deposition aligns material placement with the stress field.

**3.1.3. Microstructural and chemical composition analysis.** Edge-Focused wire arc additive manufacturing (EF-WAAM) concentrates deposition along the plate edges using a prescribed travel angle and controlled heat input per unit length, as shown in Fig 2a and Fig 4. Localizing heat at the periphery limits the volume subjected to repeated high-temperature cycles, improves heat extraction through the cooler substrate, and narrows the heat-affected zone (HAZ). Compared with conventional WAAM, this approach stabilizes bead fusion and reduces distortion when all other conditions are held constant.

Fig 13 documents the resulting macro- and microstructure for EF-WAAM. Fig 13a distinguishes three regions through the build: the top layer, the transition zone, and the base zone. In Fig 13b, the top layer shows a ferrite matrix with dispersed pearlite derived from ER70S-6 after melting and cooling. The transition zone exhibits smaller grains due to lower thermal exposure than the surface, while the base zone (CT38) is least affected by heat and retains the finest structure within the section. Across EF-WAAM, average grain size in the top and transition layers typically falls in the 5–10 μm range, which is finer than for conventional WAAM and consistent with higher flexural response under a shared specimen geometry and span.

The contrast with conventional WAAM is clear in Fig 14. Fig 14a again separates top, transition, and base regions. The top layer in Fig 17b shows coarser grains, typically 15–20 μm, reflecting greater cumulative heat input from broad-area deposition. Fig 14c indicates a less uniform transition zone with more process-related defects, including porosity and micro-cracks. Fig 14d shows the base zone similar to virgin CT38 yet more thermally affected than in EF-WAAM, indicating a modest loss of refinement. These observations—coarser grains, wider HAZ, and higher defect incidence—help explain the lower bending strengths commonly obtained for conventional WAAM when test geometry and span are identical.

Fig 12a reports EF-WAAM bending strengths from 2,414.21 to 3,338.11 MPa across the L25 trials using one three-point protocol and one specimen geometry. Fig 12b compares the best EF-WAAM case with the CT38 substrate and conventional WAAM under the same method. The stress–strain response for trial No. 7 shows high strength with useful ductility before failure, which aligns with the refined grains and cleaner fusion lines observed in Fig 13. Because three-point bending produces a nonuniform stress field with peak stress at the extreme fibers, bending strengths obtained under a fixed geometry can be higher than uniaxial tensile strengths for the same alloy; interpretation in this work therefore relies on like-for-like comparisons across groups.

From an application standpoint, the EF-WAAM peak of 3,338.11 MPa indicates potential for structures that demand high flexural capacity at low mass, including airframe members, automotive elements, and energy components subject to

complex loading. A narrower HAZ and improved fusion quality can also reduce post-processing, which supports through-put and process sustainability when path planning and interpass control are standardized.

The magnitude of the maximum value is high for a carbon-steel system when compared with typical tensile data. Within this study, the result is consistent with the measurement definition—maximum flexural stress under one fixed geometry and span—and with the microstructural evidence in Figs 13 and 14. In total, EF-WAAM specimens achieved bending strengths from 2,414.21 to 3,338.11 MPa. Fig 12a shows the distribution across the Taguchi design space, Fig 12b provides the direct comparison with the CT38 substrate and conventional WAAM, and Figs 13 and 14 demonstrate finer grains, a narrower HAZ, and fewer defects for EF-WAAM. Taken together, the data indicate that Edge-Focused deposition, coupled with disciplined control of heat input per unit length at the periphery, enhances flexural performance without changing alloy composition.

Table 5 reports chemical composition at three locations relevant to wire arc additive manufacturing (WAAM): the deposited overlay (X1, ER70S-6 feed metal), the substrate (X2, CT38 steel), and the fusion/penetration zone (X, i.e., HAZ-adjacent melt). Measurements followed ASTM E415-21 by optical emission spectrometry with uncertainty stated at the 95% confidence level. The data indicate dilution during arc fusion at the interface, and the extent of dilution differs between conventional WAAM and Edge-Focused WAAM (EF-WAAM), consistent with their different thermal footprints and heat input per unit length.

Carbon showed the expected gradation across regions. In the overlay X1, C was 0.154 wt%, matching a low-carbon filler. The substrate X2 contained 0.090 wt% C, typical of CT38. The penetration zone X registered 0.109 wt% C, intermediate between overlay and substrate, which evidences mixing during fusion. In conventional WAAM, broader heating commonly elevates penetration-zone carbon into the 0.15–0.20 wt% range; higher local C with slower cooling can increase hardness and, if uncontrolled, raise the risk of brittle constituents. In EF-WAAM, deposition at the edge restricts the fusion footprint; the penetration zone remained near 0.109 wt% C, which moderates gradients and supports ductile behavior when interpass control is maintained.

Manganese differences were pronounced. The substrate X2 contained 1.165 wt% Mn, the overlay X1 contained 0.347 wt%, and the penetration zone X was 1.035 wt%. This pattern indicates that Mn from CT38 contributes strongly in the mixed region. Because Mn acts as a deoxidizer and strengthener, large spatial swings can drive microstructural variability. Reports of conventional WAAM often show Mn variation on the order of ±0.5 wt% across the bead and HAZ, which correlates with heterogeneous fusion and lower bend performance. EF-WAAM's narrower thermal field and faster heat removal through the cooler substrate help stabilize Mn distribution across the edge melt, supporting more uniform microstructure in the transition region observed in Fig 13.

Phosphorus and sulfur remained low at all locations, consistent with low-carbon steel. Across Table 5, P ranged 0.010–0.012 wt% and S ranged 0.014–0.029 wt%. The penetration zone values were intermediate between overlay and

**Table 5. Chemical composition at different position of EF-WAAM.**

| Element | X1: Additive Material (GM70S) (%) | X: Penetration (%) | X2: Substrate (CT38) (%) |
|---|---|---|---|
| Carbon (C) | 0.154 | 0.109 | 0.090 |
| Manganese (Mn) | 0.347 | 1.035 | 1.165 |
| Phosphorus (P) | 0.012 | 0.010 | 0.011 |
| Sulfur (S) | 0.014 | 0.015 | 0.029 |
| Chromium (Cr) | 0.014 | 0.031 | 0.015 |
| Nickel (Ni) | 0.014 | 0.012 | 0.012 |

(*) Schematic illustration of chemical composition measurement positions in the EF-WAAM sample are shown in Appendix (Fig C).

substrate, indicating dilution without enrichment. In conventional WAAM, extended thermal cycles can promote local segregation of P and S in the fusion line, which increases cold-shortness risk. EF-WAAM's Edge-Focused path and controlled interpass conditions shorten exposure, limiting redistribution of P/S and helping to suppress porosity.

Chromium and nickel were present at trace levels. Nickel was approximately 0.012–0.014 wt% across locations. Chromium in the penetration zone rose to 0.031 wt%, compared with 0.014 wt% in the overlay and 0.015 wt% in the substrate, indicating modest transfer during mixing. Wider, slower thermal cycles in conventional WAAM can increase surface oxidation and reduce retained Cr at the bead surface; EF-WAAM's narrower HAZ and shorter exposure help preserve alloying near the edge and maintain consistent fusion boundaries.

Taken together, the composition profiles support the process–structure picture developed from microscopy. EF-WAAM limits the fusion area to the periphery, which reduces dilution swings in the penetration zone, narrows the HAZ, and minimizes defect precursors. The resulting chemistry is closer to a controlled blend of overlay and substrate rather than the broader, less predictable mixing typical of conventional WAAM. In conjunction with the finer grains and cleaner fusion lines observed in Fig 13 and the reduced uniformity issues in Fig 14, the Table 5 results explain the higher bending strengths measured for EF-WAAM under a fixed specimen geometry and span.

### 3.2. Prediction of experimental parameters' impact on bending strength using ANN

The ANN model not only predicts the maximum bending strength but also reconstructs the entire stress-strain curve, providing detailed insights into material stiffness (Young's modulus), ductility, and fracture point. This is particularly significant for EF-WAAM, where microstructural improvements (grain sizes of 5–10 µm compared to 15–20 µm in conventional WAAM) and a 50% reduction in the heat-affected zone result in superior mechanical performance. The ANN predictions, when combined with sensitivity analysis, will support process optimization and the design of industrial components such as aircraft frames or turbine blades, where strength and reliability are critical.

Fig 15 illustrates the architecture and training outcomes of the Artificial Neural Network (ANN) developed in this study. As shown in Fig 15a, the ANN architecture comprises six input nodes (Input layer: 6), corresponding to the process parameters of current intensity, step-over distance, travel speed, travel angle, layer thickness, and deformation. These inputs were propagated through two hidden layers (Hidden layers: 15 neurons each) with complex weights and interconnections, ultimately leading to a single output node (Output layer: 1) responsible for predicting stress. The performance and accuracy of the ANN are summarized in Fig 15b, which presents four comparative plots of target and predicted values for the training (R = 0.99482), validation (R = 0.99472), testing (R = 0.99566), and overall dataset (All, R = 0.99419). The consistently high correlation coefficients (R) demonstrate the strong predictive capability of the model. The close agreement between experimental and predicted results further confirms the robustness and reliability of the ANN in stress analysis for EF-WAAM components.

An artificial neural network (ANN) was implemented in MATLAB to predict bending strength and the full stress–strain response of EF-WAAM components. The input vector comprised six variables—current (I), step-over distance, travel angle, travel speed, layer thickness, and strain. The network used two hidden layers with 15 neurons each (sigmoid activation) and a single output neuron for bending stress (MPa), as shown in Fig 15a. The dataset contained 8,692 points aggregated from the 25 Taguchi trials in Tables 1 and 2. Data were partitioned into 70% training, 15% validation, and 15% testing to promote generalization and to monitor overfitting. Training used backpropagation with mean squared error as the loss, a learning rate of 0.01, and 1,000 epochs. Performance was high: the coefficient of determination for the training, validation, and test splits was 0.9948, 0.9947, and 0.9957, respectively (Fig 15b).

Compared with linear regression or simplified mechanical models, the ANN captured the entire curve from elastic through plastic deformation to failure [31,32]. Predicted stress–strain trajectories aligned closely with experiments for representative cases, including specimen No. 1 (2,777.15 MPa) and specimen No. 7 (3,338.11 MPa), as illustrated in Fig A1 and Fig A7 (Appendix). The average discrepancy remained under 4% across these comparisons. Similar agreement

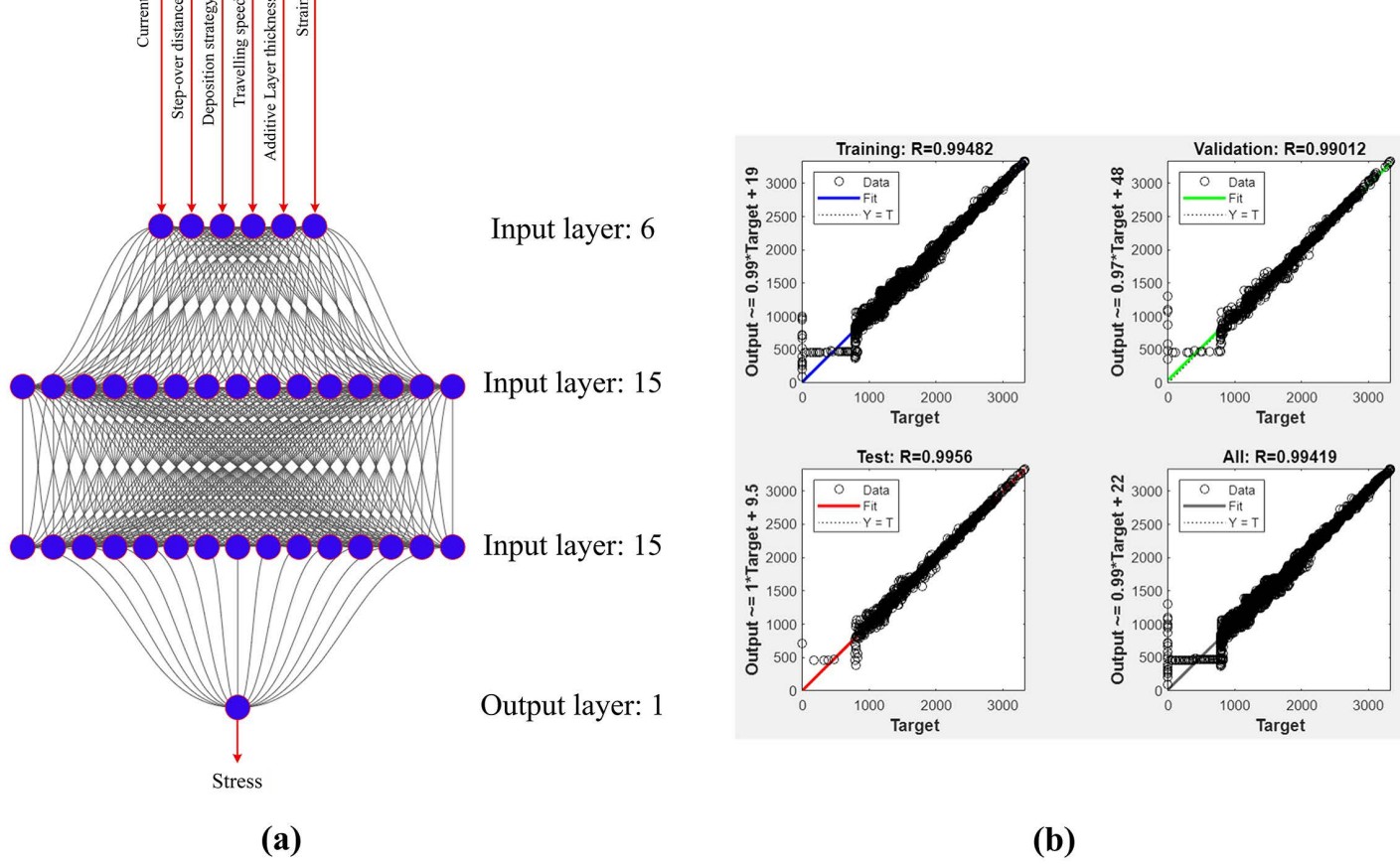

**Fig 15. The ANN structure (a) and training result of ANN (b).**

is shown for specimen No. 25 (3,169.36 MPa) in Fig A25. These results indicate that the model reproduces both peak bending strength and curve shape with sufficient fidelity for parameter screening and for assessing how changes in $I$, $a$ $α$, $F$, and $t$ shift stiffness and strength within the tested EF-WAAM window.

### 3.3. Influence of parameters on bending strength via neural network prediction

An artificial neural network (ANN) was used to quantify how process parameters influence bending strength in Edge-Focused Wire Arc Additive Manufacturing (EF-WAAM). Model inputs were current I, step-over distance a, travel angle α, travel speed F, layer thickness t, and strain ε; the output was bending stress as a function of strain. Predictions and corresponding measurements are summarized in Tables 6–8 and illustrated in Figs 16–20. The results indicate that bending performance in EF-WAAM is highly sensitive to the thermal–kinematic settings that govern heat input per unit length at the edge. Comparisons with conventional WAAM use a common specimen geometry and three-point method to isolate process effects.

Current (I) acts as a primary lever. Fig 16 shows a non-monotonic trend with a peak bending strength for specimen No. 2 at 85 A of 2,797.06 MPa and a decrease to 2,435.94 MPa at 100 A for specimen No. 5. The optimum at moderate current reflects adequate fusion without excessive arc enthalpy. Higher current extends the thermal footprint, promoting wider HAZ and increasing the likelihood of bead coarsening or defect initiation. Operating near the moderate-current window therefore stabilizes edge fusion and limits distortion while maintaining penetration.

**Table 6. Influence of welding parameters on maximum bending strength (Predicted by ANN).**

| Parameter | Value | Max Bending strength (MPa) | Note |
|---|---|---|---|
| Additive layer thickness t (mm) | 1 | 1787.77 | Small cross-section, high thermal gradient |
| | 2 | 2655.92 | Increased strength due to larger cross-section |
| | 3 | 3169.36 | Optimal strength, low thermal gradient |
| Current I (A) | 80 | 2777.15 | Stable heat input per unit length |
| | 85 | 2797.06 | Optimal strength |
| | 100 | 2435.94 | Increased thermal stress |
| Travelling speed F (mm/min) | 500 | 2663.26 | High heat exposure time |
| | 70 | 2655.62 | Balance between speed and quality |
| | 90 | 2800.01 | Optimal strength, low heat |
| Step-over distance a (mm) | 2 | 2606.14 | Uniform heat distribution |
| | 3 | 2655.00 | Stable |
| | 4 | 2476.62 | Slight heat Step-over distance |
| Travelling angle α (°) | 0 | 2775.25 | Optimal material flow |
| | 10 | 2655.19 | Stable |
| | 20 | 2414.21 | Increased internal stress |

**Table 7. Comparison of predicted (ANN) and experimental bending strength.**

| Specimen No. | Experimental Bending Strength (MPa) | Predicted Bending Strength (MPa) | Error (%) |
|---|---|---|---|
| 1 | 2777.15 | 2835.62 | 2.1 |
| 7 | 3338.11 | 3408.95 | 2.1 |
| 15 | 2414.21 | 2530.14 | 4.8 |
| 25 | 3169.36 | 3250.77 | 2.6 |

Step-over distance (a) and travel angle $\alpha$ exert smaller but measurable effects. In Fig 17, bending strength declines from 2,606.14 MPa at 2.0 mm to 2,476.62 MPa at 4.0 mm. Larger step-over reduces overlap and edge energy concentration, which can leave cold-lapped regions or under-supported flanges at the periphery. The same Fig shows a reduction from 2,775.25 MPa at $\alpha$ of 0° to 2,414.21 MPa at $\alpha$ of 20°. Larger travel angles divert heat and momentum away from the extreme fibers, lowering section efficiency in bending and broadening the local HAZ. These observations are consistent with the EF-WAAM objective of placing sound material at the outer fibers with controlled, Edge-Focused heat.

Layer thickness (t) is the dominant factor within the tested range. Fig 18 indicates an increase in bending strength from 1,787.77 MPa at t of 1.0 mm to 3,169.36 MPa at t of 3.0 mm. Thicker layers at the edge raise flange height and section modulus more quickly while keeping the number of thermal cycles per unit height lower than many thin passes. When interpass temperature is controlled, this strategy limits cumulative heating, maintains bead integrity, and reduces the incidence of fusion defects. The net effect is a larger, stiffer edge flange that carries a greater portion of the bending load without added mass away from the extreme fibers.

Travel speed (F) also contributes. Fig 20 shows an improvement from 2,663.26 MPa at 500 mm per minute to 2,800.01 MPa at 900 mm per minute. Faster travel shortens residence time, reducing heat accumulation and narrowing the edge HAZ. At the same time, wetting remains stable within the observed process window, yielding smoother overlaps and

**Table 8. Effect of edge-focussed WAAM parameters on bending strength.**

| STT | Current I (A) | Step-over distance a (mm) | Travelling angle α (°) | Travelling speed F (mm/min) | Additive layer thickness T (mm) | Prediction of Bending Stress (MPa) |
|---|---|---|---|---|---|---|
| 1 | 80 | 3 | 10 | 70 | 2 | 2777.15 |
| 2 | 85 | | | | | 2797.06 |
| 3 | 90 | | | | | 2655.22 |
| 4 | 95 | | | | | 2630.36 |
| 5 | 100 | | | | | 2435.94 |
| 6 | 90 | 2 | | | | 2386.05 |
| 7 | | 2.5 | | | | 2606.14 |
| 8 | | 3 | | | | 2655.00 |
| 9 | | 3.5 | | | | 2647.05 |
| 10 | | 4 | | | | 2476.62 |
| 11 | | 3 | 0 | | | 2775.25 |
| 12 | | | 5 | | | 2722.57 |
| 13 | | | 10 | | | 2655.19 |
| 14 | | | 15 | | | 2595.36 |
| 15 | | | 20 | | | 2414.21 |
| 16 | | | 10 | 500 | | 2663.26 |
| 17 | | | | 600 | | 2679.45 |
| 18 | | | | 700 | | 2655.62 |
| 19 | | | | 800 | | 2709.97 |
| 20 | | | | 900 | | 2800.01 |
| 21 | | | | 70 | 1 | 1787.77 |
| 22 | | | | | 1.5 | 2242.37 |
| 23 | | | | | 2 | 2655.92 |
| 24 | | | | | 2.5 | 2938.24 |
| 25 | | | | | 3 | 3169.36 |

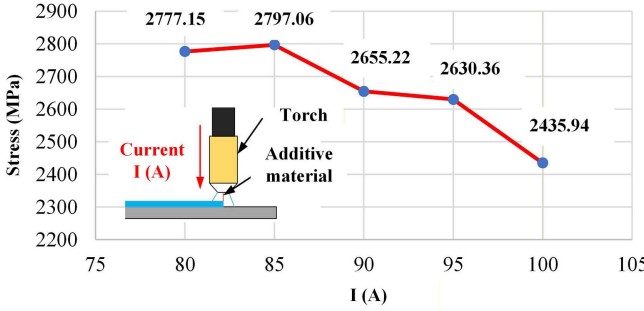

**Fig 16. Effect of current on the bending strength of Edge-Focused WAAM.**

fewer reheated segments than at lower speeds. This balance of reduced thermal exposure and sound fusion aligns with the microstructural differences reported for EF-WAAM in Figs 14 and 15.

Taken together, Tables 6–8 and Figs 16–20 support the following parameter guidance for EF-WAAM under the fixed specimen geometry used here: moderate current to avoid excess heat; small step-over and small travel angle to

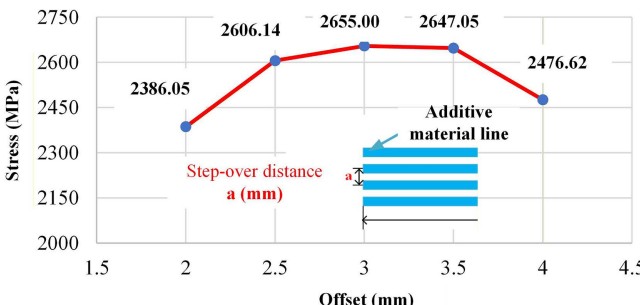

**Fig 17. Effect of Step-over distance on the bending strength of Edge-Focused WAAM.**

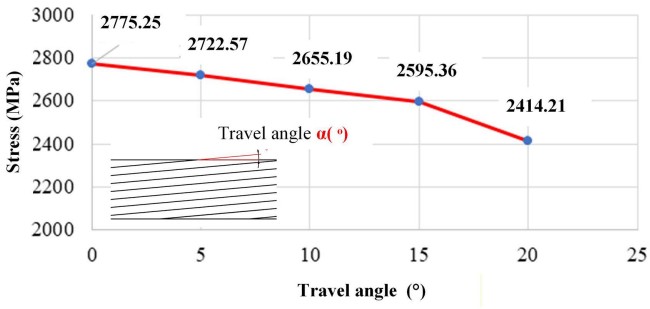

**Fig 18. Effect of Travel angle on the bending strength of Edge-Focused WAAM.**

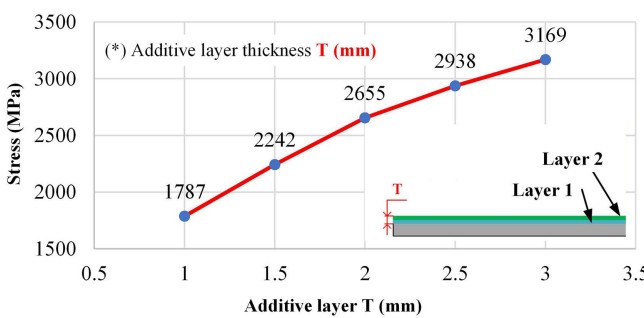

**Fig 19. Effect of Additive layer thickness on the bending strength of Edge-Focused WAAM.**

concentrate energy and material at the periphery; larger layer thickness within the stable fusion window to increase section efficiency with fewer thermal cycles; and higher travel speed within the wetting-stable regime to limit heat input per unit length. The ANN reproduces both peak bending strength and curve shape across the Taguchi space, providing a practical tool to screen [$I, a, \alpha, F, t$] settings before fabrication and to prioritize Edge-Focused conditions that improve flexural capacity without increasing mass.

Table 7 compares artificial neural network (ANN) predictions with measured bending strength for the EF-WAAM data-set under one specimen geometry and a single three-point protocol. The absolute percentage error ranged from 2.1% for specimens No. 1 and No. 7 to 4.8% for No. 15. These errors are lower than the 10–15% dispersions commonly

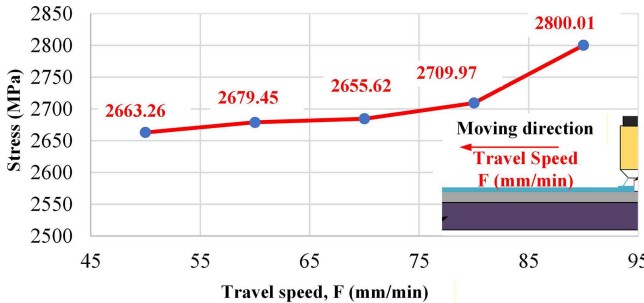

**Fig 20. Effect of travel speed on the bending strength of Edge-Focused WAAM.**

reported for linear regression surrogates in prior WAAM studies [32], indicating that the ANN captured the main nonlinear effects of current (I), step-over distance (a), travel angle α\alphaα, travel speed (F), and layer thickness (t) within the tested window.

Two factors likely contribute to the reduced error. First, the ANN was trained on curve-level data rather than single peak values, which helps the model learn elastic–plastic trends and the approach to failure, improving generalization across the Taguchi cases. Second, the Edge-Focused WAAM (EF-WAAM) workflow produced more uniform beads and a narrower heat-affected zone (HAZ) than broad-area deposition, leading to fewer porosity sites and cleaner fusion boundaries. Lower microstructural noise at the edge reduces run-to-run variability, which improves the alignment between predictions and measurements. Together, these conditions yield prediction errors in Table 7 that are small enough for process screening and for selecting parameter sets that raise flexural performance without changing alloy composition or specimen geometry.

### 3.4. Application of Edge-Focused wire arc additive manufacturing in forming 3D spiral channel

Fig 21 outlines a practical workflow to build 3D spiral channels using Edge-Focused wire arc additive manufacturing (EF-WAAM). The process starts with 3D channel design (Fig 24a), where helix geometry, wall thickness, and curvature

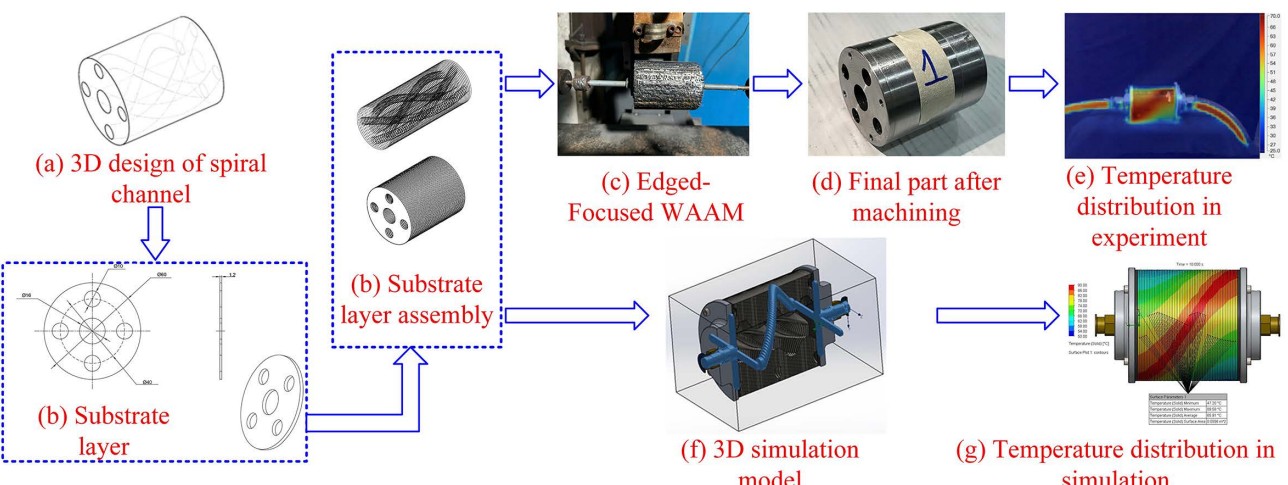

**Fig 21. Process of edge-focused wire arc additive manufacturing in forming 3D spiral channel.**

limits are set to meet tolerance and tool-access constraints. The substrate is then prepared and assembled (Fig 24b): plates are profiled, aligned on a rigid fixture, and referenced with datums to stabilize the base before deposition. This step minimizes gap variation and provides a consistent heat sink, which is critical for dimensional control in curved features.

Material addition follows an Edge-Focused path plan (Fig 24c). The torch tracks the outer bounds of the helical channel with a prescribed travel angle and a travel speed chosen to manage heat input per unit length at the periphery. Deposition along the edge concentrates material where bending and pressure loads are carried, while the cooler bulk substrate extracts heat quickly, narrowing the heat-affected zone (HAZ) and reducing distortion. Several edge layers establish the channel walls; intermittent skimming restores a flat datum for the next passes. After reaching near-net geometry, finish machining defines the groove depth and surface quality (Fig 24d), leaving a smooth flow path and tight dimensional tolerance.

Thermal behavior is documented by measurement and simulation to de-risk the build. The experimental temperature map (Fig 24e) shows localized, bounded gradients around the edge path, consistent with rapid heat removal through the substrate. The 3D simulation model (Fig 24f) and the corresponding simulated temperature distribution (Fig 24g) predict the same concentration of heat at the periphery and guide parameter choices—current, travel angle, travel speed, inter-pass limits—before fabrication. Using both views enables adjustment of dwell and bead overlap at tight radii of the helix, where thermal accumulation and waviness are most likely.

This combined design–build–verify approach addresses the typical difficulties of helical and internal channels in metal AM: maintaining wall straightness on a curved path, limiting HAZ growth, and achieving uniform fusion around changing curvature. EF-WAAM places metal only where it adds structural and functional value, keeps the part close to net shape throughout the loop, and reduces post-processing length and variability. The workflow in Fig 24 is transferable to other WAAM variants by retaining the same measurement gates and analysis blocks while adapting the edge-first path plan and finishing allowances to the target channel geometry.

## 4. Conclusion

This study demonstrates that Edge-Focused Wire Arc Additive Manufacturing (EF-WAAM) improves mechanical response and build quality for CT38 steel using ER70S-6 filler when compared with conventional WAAM and the substrate baseline. Concentrating deposition at the edges reduces the volume repeatedly exposed to high temperature, narrows the heat-affected zone (HAZ), and limits residual distortion. Under a single three-point method and one specimen geometry, EF-WAAM coupons reached bending strengths from 2,414.21 to 3,338.11 MPa; the peak value represents about 171% gain over the CT38 baseline of 1,231.5 MPa and exceeds typical conventional WAAM values below 2,000 MPa under comparable geometry. A mass-specific performance index P of 420.4 MPa·cm³/g was recorded for the best EF-WAAM case, which corresponds to increases of 40.4% relative to the substrate and 83.3% relative to conventional WAAM. These gains reflect section-efficient material placement at the extreme fibers and controlled heat input per unit length at the edge.

Parameter effects resolved by the artificial neural network (ANN) are consistent with the process physics. Layer thickness t dominated within the tested range, with bending strength rising from 1,787.77 MPa at 1.0 mm to 3,169.36 MPa at 3.0 mm, indicating faster growth of the edge flange and fewer thermal cycles per unit height when interpass limits are respected. Travel speed F contributed positively, improving from 2,663.26 MPa at 500 mm/min to 2,800.01 MPa at 900 mm/min, which shortened residence time and reduced heat accumulation at the periphery. Current I showed an optimum near 85 A with 2,797.06 MPa for specimen No. 2 and a drop to 2,435.94 MPa at 100 A (specimen No. 5), consistent with adequate fusion at moderate current and coarsening or defect initiation at higher arc enthalpy. Step-over a and travel angle α produced smaller, monotonic reductions across their ranges, reflecting diminished edge energy concentration and lower section efficiency when overlap is reduced or the torch is inclined away from the extreme fibers. The ANN reproduced both peak strength and curve shape across the Taguchi space with errors below five percent on representative cases, supporting use for S/N-guided screening of [I,a,α,F,t] before fabrication.

The workflow extended to a 3D spiral channel demonstrator. EF-WAAM toolpaths followed the helical edges, and finishing established the final groove geometry. Temperature mapping and finite-element simulation showed localized, bounded gradients at the edge, aligning with the observed narrower HAZ and reduced waviness. Together, the mechanical results, composition and microstructure evidence, and ANN predictions indicate that EF-WAAM provides higher flexural capacity and better mass efficiency at constant density, with a transferable process that supports complex features when path planning, interpass control, and measurement gates are kept consistent.

Future work will extend the EF-WAAM strategy to curved and closed-loop geometries by leveraging CNC/robotic path planning to reinforce non-linear edges and round profiles. Additional studies will focus on interfacial integrity under multi-axial loading and the influence of curvature on heat localization and residual stress evolution. These developments aim to translate the demonstrated edge-focused concept into complex, application-relevant geometries.

## Supporting information

**S1 Fig. Case 1: This is the comparation between prediction and experimental of stress-strain curves of case 1.**
(JPG)

**S2 Fig. Case 2: This is the comparation between prediction and experimental of stress-strain curves of case 2.**
(JPG)

**S3 Fig. Case 3: This is the comparation between prediction and experimental of stress-strain curves of case 3.**
(JPG)

**S4 Fig. Case 4: This is the comparation between prediction and experimental of stress-strain curves of case 4.**
(JPG)

**S5 Fig. Case 5: This is the comparation between prediction and experimental of stress-strain curves of case 5.**
(JPG)

**S6 Fig. Case 6: This is the comparation between prediction and experimental of stress-strain curves of case 6.**
(JPG)

**S7 Fig. Case 7: This is the comparation between prediction and experimental of stress-strain curves of case 7.**
(JPG)

**S8 Fig. Case 8: This is the comparation between prediction and experimental of stress-strain curves of case 8.**
(JPG)

**S9 Fig. Case 9: This is the comparation between prediction and experimental of stress-strain curves of case 9.**
(JPG)

**S10 Fig. Case 10: This is the comparation between prediction and experimental of stress-strain curves of case 10.**
(JPG)

**S11 Fig. Case 11: This is the comparation between prediction and experimental of stress-strain curves of case 11.**
(JPG)

**S12 Fig. Case 12: This is the comparation between prediction and experimental of stress-strain curves of case 12.**
(JPG)

**S13 Fig. Case 13: This is the comparation between prediction and experimental of stress-strain curves of case 13.**
(JPG)

**S14 Fig. Case 14: This is the comparation between prediction and experimental of stress-strain curves of case 14.** (JPG)

**S15 Fig. Case 15: This is the comparation between prediction and experimental of stress-strain curves of case 15.** (JPG)

**S16 Fig. Case 16: This is the comparation between prediction and experimental of stress-strain curves of case 16.** (JPG)

**S17 Fig. Case 17: This is the comparation between prediction and experimental of stress-strain curves of case 17.** (JPG)

**S18 Fig. Case 18: This is the comparation between prediction and experimental of stress-strain curves of case 18.** (JPG)

**S19 Fig. Case 19: This is the comparation between prediction and experimental of stress-strain curves of case 19.** (JPG)

**S20 Fig. Case 20: This is the comparation between prediction and experimental of stress-strain curves of case 20.** (JPG)

**S21 Fig. Case 21: This is the comparation between prediction and experimental of stress-strain curves of case 21.** (JPG)

**S22 Fig. Case 22: This is the comparation between prediction and experimental of stress-strain curves of case 22.** (JPG)

**S23 Fig. Case 23: This is the comparation between prediction and experimental of stress-strain curves of case 23.** (JPG)

**S24 Fig. Case 24: This is the comparation between prediction and experimental of stress-strain curves of case 24.** (JPG)

**S25 Fig. Case 25: This is the comparation between prediction and experimental of stress-strain curves of case 25.** (JPG)

**S26 Fig. Main effects plot for S/N ratios of flexural strength of EF-WAAM specimens.** This is a main effects plot showing that layer thickness has the strongest influence on the S/N ratio of flexural strength in EF-WAAM specimens. (JPG)

**S27 Fig. Schematic illustration of chemical composition measurement positions in the EF-WAAM sample.** This is a schematic illustration showing the chemical composition measurement positions in the EF-WAAM sample, including the additive material, penetration area, and substrate. (JPG)

**S28 Fig. Graphical Abstract.** (JPG)

## Acknowledgments

The authors acknowledge the support of School of Industrial Engineering and Management, International University, Vietnam National University HCMC, Vietnam and HCMC University of Technology and Engineering, Vietnam for this research.

## Author contributions

**Conceptualization:** Tran Le Hong Ngoc, Pham Son Minh.

**Data curation:** Pham Son Minh.

**Formal analysis:** Pham Son Minh.

**Funding acquisition:** Pham Son Minh.

**Investigation:** Tran Le Hong Ngoc, Pham Son Minh.

**Methodology:** Tran Le Hong Ngoc, Pham Son Minh.

**Project administration:** Pham Son Minh.

**Supervision:** Tran Le Hong Ngoc, Ha Thi Xuan Chi, Van-Thuc Nguyen, Pham Son Minh.

**Validation:** Tran Le Hong Ngoc, Ha Thi Xuan Chi, Van-Thuc Nguyen, Pham Son Minh.

**Visualization:** Tran Le Hong Ngoc, Ha Thi Xuan Chi, Van-Thuc Nguyen, Pham Son Minh.

**Writing – original draft:** Tran Le Hong Ngoc, Pham Son Minh.

**Writing – review & editing:** Tran Le Hong Ngoc, Pham Son Minh.

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
