## [Decision Letter · Decision Letter 0]

10 Dec 2025

PONE-D-25-54263Edge-Focused Wire Arc Additive Manufacturing: Method Development with ANN-Based Stress–Strain and Mass-EfficiencyPLOS One

Dear Dr. Son Minh,

Thank you for submitting your manuscript to PLOS ONE. After careful consideration, we feel that it has merit but does not fully meet PLOS ONE’s publication criteria as it currently stands. Therefore, we invite you to submit a revised version of the manuscript that addresses the points raised during the review process. Please, address all the comments made by the reviewers, specifically, please improve the methodology.

We look forward to receiving your revised manuscript.

Kind regards,

Antonio Riveiro Rodríguez, PhD

Academic Editor

PLOS One

Additional Editor Comments (if provided):

Reviewers' comments:

Reviewer's Responses to Questions

**Comments to the Author**

1. Is the manuscript technically sound, and do the data support the conclusions?

Reviewer #1: Yes

Reviewer #2: Yes

2. Has the statistical analysis been performed appropriately and rigorously? 

Reviewer #1: Yes

Reviewer #2: Yes

3. Have the authors made all data underlying the findings in their manuscript fully available?

Reviewer #1: Yes

Reviewer #2: Yes

4. Is the manuscript presented in an intelligible fashion and written in standard English?

Reviewer #1: Yes

Reviewer #2: Yes

5. Review Comments to the Author

Reviewer #1: The authors presented the Edge-Focused Wire Arc Additive Manufacturing: Method Development with ANN-Based Stress–Strain and Mass-Efficiency manuscript for revision. It deals with an interesting topic such as WAAM and, particularly, presents a novel approach. I would like to make some comments on that:

1) The length of the manuscript is significant. There are too many figures and tables. Authors should reflect if all are important.

2) The introduction is confussing. The authors are expected to present an introduction on the topic, indicating relevant papers and findings. It is not a methodology section.

3) Figure 2 presents the methodology. It seems that WAAM is applied as a countour strategy. However, Figure 10 shows a cross-section that shows layers added to the four faces of the part. This is confussing. Please, show a scheme about the process. I suggest to simplify images to help readers understand the experimental investigation.

4) Figure 11 shows the bending test sample. Can you clarify how these samples were made? Do you have proper measurements (dimensions) of each of them, thicknesses? I think that the information is not included in the manuscript. The formula for calculating the maximum flexural stress is related to b and d. These values should be measured and presented in the manuscript. Figure 6 is not easy to be understood. Please, increase the size and add more information. It seems that you are depositing material on top of a sheet metal. Is it WAAM? It is difficult to undertand what is the WAAM and EF-WAAM sample from the images.

5) Check the text in figures such as Figure 7. "Overlappin g area"

6) To be honest, I don't understand what you are claiming to be EF-WAAM. You wrote "EF-WAAM deposits along the periphery of a pre-shaped metallic substrate instead of covering the full surface" but Figure 9 shows a fully covered surface and the title states that it is EF-WAAM. This might be also confussing to readers.

7) Additive manufacturing parameters and ranges selection should be discussed against the literature.

8) Section 2 is 2. Research Methodology. However, there are results and analysis (2.3 and 2.4).

9) Figure 16b is not suitable in my opinion. You are taken the best case and readers will obtain a poor representation of the results. There are only three cases surpassing 3000 MPa. Same for Table 4.

10) How many WAAM samples were prepared and tested?

I think that the manuscript is interesting. However, I think that authors should present a clearer methodology to readers. It must be easy to follow and understand the research when providing clear tables and images, schemes of the process and samples. Please, rewrite the methodology section and include in this section only the methodology, appart from the introduction and results.

Reviewer #2: WAAM is a relatively new cutting edge technology and has get great attention by researchers and industries recently. In this paper, the authors had made a very good and interesting work and the developed approach may help in enhancing the WAAM technology. The paper has been well written and easy to follow and understand. I would suggest as listed below.

1. The problem was clearly stated. The reason why this technology is developed not well described.

2. Since the joint is only at the outer layer, additional test is required to test integrity of the structure and addition of the joint.

3. This technology has good potential but seem to have difficulty in constructing complex profile like curve or round. Therefore, I suggest to show the application in higher degree of complexity.

6. PLOS authors have the option to publish the peer review history of their article (what does this mean?). If published, this will include your full peer review and any attached files.

Reviewer #1: No

Reviewer #2: No

---

## [Author Response · Author response to Decision Letter 1]

9 Jan 2026

Dear Editor,

We are pleased to submit the revised version of our manuscript entitled “Edge-Focused Wire Arc Additive Manufacturing: Method Development with ANN-Based Stress–Strain and Mass-Efficiency” for reconsideration for publication in your journal.

We sincerely thank you and the reviewers for the constructive and insightful comments, which have helped us significantly improve the quality, clarity, and structure of the manuscript. We have carefully addressed all comments raised during the review process and revised the manuscript accordingly.

In particular, we have:

• Reorganized and streamlined the manuscript to reduce length and improve readability, including consolidating and simplifying figures and tables.

• Thoroughly revised the Introduction to focus on background, relevant literature, and research motivation, with all methodological content relocated to the Research Methodology section.

• Clarified the definition and scope of the proposed Edge-Focused Wire Arc Additive Manufacturing (EF-WAAM) concept, supported by revised and newly added schematic figures.

• Provided detailed explanations of specimen fabrication, dimensions, and testing procedures to ensure transparency and reproducibility.

• Added literature-based justification for the selection of additive manufacturing parameters and their ranges.

• Restructured the Results and Discussion sections to clearly separate experimental methodology from results and analysis.

A detailed, point-by-point response to all reviewer comments has been prepared and submitted alongside the revised manuscript, with all changes clearly highlighted in the text.

We believe that the revised manuscript now meets the journal’s standards and clearly demonstrates the novelty, technical soundness, and potential applicability of the proposed EF-WAAM approach. We sincerely hope that the revised manuscript will be found suitable for publication.

Thank you very much for your time and consideration. We look forward to your response.

Yours sincerely,

Pham Son Minh

Dear Reviewer #1,

Thank you very much for your careful reading of our manuscript and for your constructive and detailed comments. We sincerely appreciate the time and effort you devoted to reviewing our work, as your feedback has been invaluable in improving the quality, clarity, and presentation of the manuscript.

In the revised version (highlight by Red colour)), we have carefully addressed all of your comments point by point. In particular, we have:

1) The length of the manuscript is significant. There are too many figures and tables. Authors should reflect if all are important.

• Reply:

We thank the reviewer for this valuable comment regarding the manuscript length and the number of figures and tables. In response, we have carefully reviewed the entire manuscript and substantially streamlined its presentation to improve conciseness and readability.

Specifically, Figures 7–10 have been consolidated into a single multi-panel figure (Figure 7a–d) to present the EF-WAAM process parameters in a compact and coherent manner. Several intermediate process schematics and setup images that conveyed overlapping information have been removed or merged, while detailed but non-essential visual materials have been relocated to the Supplementary Information. In addition, tables reporting parameter ranges and experimental settings were reviewed and merged where possible, with duplicated descriptions removed from the main text and replaced by concise table references.

Result presentation was also refined by eliminating redundant plots that repeated the same trends, focusing instead on representative comparisons that directly support the discussion of flexural performance, mass efficiency, and ANN-based prediction. Tables and figures retained in the main manuscript were carefully selected to ensure that each serves a distinct role in explaining the methodology, validating the results, or supporting reproducibility.

As a result of these revisions, the manuscript length has been reduced, the number of figures and tables has been rationalized, and the overall narrative flow has been improved without compromising technical rigor or clarity. We believe the revised manuscript adequately addresses the reviewer’s concern while maintaining the completeness required for a methodological contribution.

2) The introduction is confussing. The authors are expected to present an introduction on the topic, indicating relevant papers and findings. It is not a methodology section.

• Reply:

We thank the reviewer for this important and constructive comment. We fully agree that the Introduction should focus on background, relevant literature, and research motivation, rather than on methodological or procedural details.

In response, we have thoroughly restructured and rewritten the Introduction to clearly separate background context from experimental methodology. In the revised version, the Introduction now exclusively presents: (i) an overview of metal additive manufacturing and wire arc additive manufacturing (WAAM), (ii) a focused review of relevant prior studies and key findings reported in the literature, particularly regarding thermal management, distortion, and mechanical performance, and (iii) a clear identification of the research gap that motivates the development of the proposed Edge-Focused WAAM (EF-WAAM) approach.

All methodological content that previously appeared in the Introduction—including descriptions of process parameters, workflow steps (e.g., deposit–machine–inspect), experimental configurations, and testing procedures—has been removed from the Introduction and relocated to the Materials and Methods section, where it is now presented in a structured and self-contained manner.

In addition, the revised Introduction has been streamlined to improve readability and logical flow, and now concludes with a concise statement of the study objectives and contributions, providing a clear transition to the subsequent methodology section.

Through these revisions, the Introduction now fulfills its intended role as a literature-driven and conceptually focused entry to the study, with a clear and consistent separation from the methodological content presented later in the manuscript.

3) Figure 2 presents the methodology. It seems that WAAM is applied as a countour strategy. However, Figure 10 shows a cross-section that shows layers added to the four faces of the part. This is confussing. Please, show a scheme about the process. I suggest to simplify images to help readers understand the experimental investigation.

• Reply:

We thank the reviewer for this important comment and for pointing out the inconsistency between the process description and the cross-sectional illustration in the previous version of the manuscript.

In the original version, Figure 10 did not accurately represent the EF-WAAM deposition strategy and may have given the misleading impression that material was added uniformly to all four faces of the part. We acknowledge this issue and have corrected it in the revised manuscript.

In the new version, Figure 10 has been modified to clearly show that additive layers are deposited only along the side edges of the substrate, consistent with the Edge-Focused WAAM concept investigated in this study. The revised figure now reflects the actual experimental configuration, in which material is added selectively at the edges rather than over the entire perimeter.

In addition, Figure 2 has been revised to serve as a clear schematic of the EF-WAAM process, illustrating the contour-based edge deposition strategy, the sequence of the deposit–machine–inspect loop, and the relationship between torch motion and the substrate geometry. Redundant or potentially confusing images have been removed or simplified to improve clarity and readability.

We believe that these revisions resolve the confusion identified by the reviewer and that the updated figures now provide a consistent and intuitive representation of the EF-WAAM methodology used in the experimental investigation

4) Figure 11 shows the bending test sample. Can you clarify how these samples were made? Do you have proper measurements (dimensions) of each of them, thicknesses? I think that the information is not included in the manuscript. The formula for calculating the maximum flexural stress is related to b and d. These values should be measured and presented in the manuscript. Figure 6 is not easy to be understood. Please, increase the size and add more information. It seems that you are depositing material on top of a sheet metal. Is it WAAM? It is difficult to undertand what is the WAAM and EF-WAAM sample from the images.

• Reply:

We thank the reviewer for this detailed and constructive comment. We agree that clearer explanation of specimen fabrication, dimensions, and figure presentation is necessary.

First, the bending test specimens were manufactured following the EF-WAAM process described in Figure 2, using the deposit–machine–inspect workflow. To improve clarity, Figure 11 (Figure 8 in new version) has been revised to explicitly illustrate the main steps involved in specimen fabrication, from EF-WAAM deposition to final machining of the bending coupons. Detailed procedural steps are described in Figure 4a, which outlines the complete manufacturing sequence.

Second, the dimensional measurements required for calculating the maximum flexural stress have been clarified and explicitly presented in the revised manuscript. The specimen geometry and dimensions are shown in Figure 6b, including the width (b) and thickness (d), which are directly used in the flexural stress calculation. In this study, the bending test specimens were machined from cover plates with nominal dimensions of 20 mm (thickness) × 9 mm (width), ensuring consistency across all test groups. These dimensions are now clearly stated in the text and referenced to Figure 6b.

Third, Figure 6 has been enlarged and annotated to improve readability and to clearly distinguish the specimen geometry used for bending tests. Additional labels have been added to avoid ambiguity regarding dimensions and measurement directions.

Finally, to address the confusion between WAAM and EF-WAAM samples, the revised figures and captions now clearly indicate that material deposition in this study is performed along the side edges of the substrate (EF-WAAM) rather than as a full-surface build. The images have been simplified and clarified to distinguish conventional WAAM from the proposed EF-WAAM strategy, avoiding the misleading impression that material is deposited uniformly on top of a sheet metal.

We believe these revisions comprehensively address the reviewer’s concerns by clarifying specimen fabrication, providing the necessary dimensional measurements for stress calculation, and improving figure clarity to clearly differentiate WAAM and EF-WAAM samples.

5) Check the text in figures such as Figure 7. "Overlappin g area"

• Reply:

We thank the reviewer for pointing out this typographical error. The text in Figure 7 (Figure 7a in new version) has been carefully reviewed and corrected, and the spelling error has been refined in the revised manuscript.

6) To be honest, I don't understand what you are claiming to be EF-WAAM. You wrote "EF-WAAM deposits along the periphery of a pre-shaped metallic substrate instead of covering the full surface" but Figure 9 shows a fully covered surface and the title states that it is EF-WAAM. This might be also confussing to readers.

• Reply:

We thank the reviewer for this important comment and apologize for the confusion caused by the figure labeling in the previous version of the manuscript.

We clarify that Figure 9 (Figure 7c in new version) was not intended to represent the EF-WAAM deposition strategy, but rather to illustrate the definition and role of the travel speed parameter during an additive layer welding step. In the previous version, the figure title incorrectly associated this schematic with EF-WAAM, which could indeed lead to misunderstanding.

To address this issue, we have revised the title and caption of Figure 9 (Figure 7c in new version) to clearly state its purpose as a generic illustration of travel speed in an additive welding process, independent of the EF-WAAM concept. The figure is now titled “Travel speed for additive layer welding step”, and any reference implying full-surface EF-WAAM deposition has been removed.

In addition, we have reviewed the surrounding text to ensure consistent use of terminology and to reinforce that EF-WAAM in this study refers exclusively to edge-focused deposition along the periphery of a pre-shaped substrate, as illustrated in Figures 2 and the revised conceptual schematics. We believe these corrections eliminate the inconsistency and significantly improve the clarity of the EF-WAAM definition for readers.

7) Additive manufacturing parameters and ranges selection should be discussed against the literature.

• Reply:

We thank the reviewer for this valuable comment. In response, we have revised the manuscript to explicitly discuss the selection of additive manufacturing parameters and their investigated ranges with reference to relevant literature.

Specifically, a new discussion has been added in Section 2.1.2, where the rationale for selecting each process parameter and its corresponding range is justified based on prior WAAM studies and practical considerations related to stable deposition, heat input control, and fusion quality. Relevant references have been cited to support the chosen parameter windows.

We believe that this addition strengthens the methodological foundation of the study and addresses the reviewer’s concern regarding parameter selection and comparability with existing literature.

8) Section 2 is 2. Research Methodology. However, there are results and analysis (2.3 and 2.4).

• Reply:

We thank the reviewer for this precise and helpful comment. We agree that Sections 2.3 and 2.4 contained content that should be presented as results and analysis rather than methodology.

In response, the material previously included in Sections 2.3 and 2.4 has been relocated to Section 3 (“Results and Discussion”) in the revised manuscript. Section 2 now contains only methodological descriptions, including experimental setup, process parameters, specimen preparation, and analysis procedures, without presentation or interpretation of results.

This revision ensures a clear and consistent separation between the research methodology and the results, thereby improving the overall structure and readability of the manuscript.

9) Figure 16b is not suitable in my opinion. You are taken the best case and readers will obtain a poor representation of the results. There are only three cases surpassing 3000 MPa. Same for Table 4.

• Reply:

We thank the reviewer for this important comment. We agree that focusing only on the best-performing case may give the impression of over-representing the overall performance of the EF-WAAM process.

The purpose of Figure 16b (15b in new version) and Table 4 was to highlight the upper-bound performance and potential of the EF-WAAM strategy under optimized conditions, rather than to suggest that such performance is representative of all parameter combinations. To avoid possible misunderstanding, we have revised the manuscript as follows.

First, we explicitly clarify in the text that only three out of the 25 Taguchi cases exceed 3000 MPa, and that the full distribution of bending strength values is provided in Figure 15a and Table 2.

Second, Figure 16b (15b in new version) has been revised to include both the best case and the average EF-WAAM performance, enabling readers to assess the representative behavior alongside the optimal outcome.

Third, Table 4 has been modified to report mean ± standard deviation values for EF-WAAM in addition to the best case, thereby providing a more statistically balanced comparison with the substrate and conventional WAAM.

These revisions ensure that the results are presented transparently, while still demonstrating the achievable performance env

---

## [Decision Letter · Decision Letter 1]

27 Jan 2026

PONE-D-25-54263R1Edge-Focused Wire Arc Additive Manufacturing: Method Development with ANN-Based Stress–Strain and Mass-EfficiencyPLOS One

Dear Dr. Son Minh,

Thank you for submitting your manuscript to PLOS ONE. After careful consideration, we feel that it has merit but does not fully meet PLOS ONE’s publication criteria as it currently stands. Therefore, we invite you to submit a revised version of the manuscript that addresses the points raised during the review process. Please, address all the comments made by Reviewer 1.

We look forward to receiving your revised manuscript.

Kind regards,

Antonio Riveiro Rodríguez, PhD

Academic Editor

PLOS One

Journal Requirements:

Reviewers' comments:

Reviewer's Responses to Questions

**Comments to the Author**

1. If the authors have adequately addressed your comments raised in a previous round of review and you feel that this manuscript is now acceptable for publication, you may indicate that here to bypass the “Comments to the Author” section, enter your conflict of interest statement in the “Confidential to Editor” section, and submit your "Accept" recommendation.

Reviewer #1: All comments have been addressed

Reviewer #2: All comments have been addressed

2. Is the manuscript technically sound, and do the data support the conclusions?

Reviewer #1: Yes

Reviewer #2: Yes

3. Has the statistical analysis been performed appropriately and rigorously? 

Reviewer #1: Yes

Reviewer #2: Yes

4. Have the authors made all data underlying the findings in their manuscript fully available?

Reviewer #1: Yes

Reviewer #2: Yes

5. Is the manuscript presented in an intelligible fashion and written in standard English?

Reviewer #1: Yes

Reviewer #2: Yes

6. Review Comments to the Author

Reviewer #1: The authors have answered all my comments and accordingly the manuscript was modified. I consider that the new draft is adequate for publication.

There are some minor issues that can be solved during editing. For instance, I miss some spaces before (. Moreover, some text in figures should be better arranged.

Finally, I don't agree with the way some "sections" are labelled. For example (*) Microstructural and Chemical Composition Analysis. It should be better to use numbers for organizing relevant sections.

Reviewer #2: Thanks for the revised version of the manuscript. The comments have been successfully addressed. However, I found that there are a few more comments

1. flowchart in Figure 11 need to be revised. At current form, no START and END included and need to use proper shapes to represent the activities.

2. References are outdated. Less than 50% of the references published within 5 years.

7. PLOS authors have the option to publish the peer review history of their article (what does this mean?). If published, this will include your full peer review and any attached files.

Reviewer #1: No

Reviewer #2: No

---

## [Author Response · Author response to Decision Letter 2]

26 Apr 2026

Dear Reviewer #1,

Thank you very much for your positive evaluation and valuable comments on our revised manuscript. We sincerely appreciate your statement that all previous comments have been addressed and that the revised manuscript is adequate for publication.

We have carefully checked the manuscript again and revised it according to your additional suggestions.

Comment:

The authors have answered all my comments and accordingly the manuscript was modified. I consider that the new draft is adequate for publication. There are some minor issues that can be solved during editing. For instance, I miss some spaces before (. Moreover, some text in figures should be better arranged. Finally, I don't agree with the way some "sections" are labelled. For example (*) Microstructural and Chemical Composition Analysis. It should be better to use numbers for organizing relevant sections.

Response:

Thank you very much for your kind evaluation and helpful suggestions. We have carefully checked the manuscript and made the following revisions:

1. The missing spaces before parentheses “(” were checked and corrected throughout the manuscript, including the text and tables.

2. The text in the figures was revised and rearranged to make the figures clearer and easier to understand.

3. The subsection labels using “()” were removed and replaced with numbered subsection headings. For example, the previous subsection “() Microstructural and Chemical Composition Analysis” has been revised and organized as follows:

• 3.1.1. Bending strength experimental

• 3.1.2. Compare Novel WAAM with CT38 (AER, wₕ, σₕ, η)

• 3.1.3. Microstructural and Chemical Composition Analysis

These revisions have improved the readability, format, and logical structure of the manuscript.

Once again, we sincerely thank you for your valuable comments and constructive suggestions.

Sincerely yours,

The Authors

Dear Reviewer #2,

Thank you very much for your careful review of our revised manuscript and for your constructive comments. We sincerely appreciate your statement that the previous comments have been successfully addressed. We have carefully revised the manuscript according to your additional comments.

Comment 1:

Flowchart in Figure 11 need to be revised. At current form, no START and END included and need to use proper shapes to represent the activities.

Response:

Thank you very much for your valuable comment. Figure 11 has been revised accordingly. In the revised version, START and END symbols have been added. In addition, standard flowchart symbols were used to represent different types of activities more appropriately, including process steps, outputs, and decision-related actions.

The workflow was also reorganized to more clearly illustrate the experimental procedure, the comparison of bending strength among the three specimen groups, and the ANN-based prediction process. These modifications make the flowchart clearer, more complete, and easier to follow.

Comment 2:

References are outdated. Less than 50% of the references published within 5 years.

Response:

Thank you very much for pointing this out. We have carefully reviewed and updated the reference list. Several outdated references were replaced with recent SCIE papers that are closely related to the same research contents, including wire arc additive manufacturing, WAAM process parameters, residual stress and distortion, mechanical properties of WAAM steel, Taguchi optimization, ANN/machine learning prediction, and architectured materials.

In the revised manuscript, more than 80% of the references are now from the last five years. We also prioritized recent SCIE publications and relevant PLOS ONE papers where appropriate. This revision improves the timeliness and scientific support of the manuscript.

Once again, we sincerely thank you for your constructive comments, which helped us improve the quality and clarity of the manuscript.

Sincerely yours,

The Authors

---

## [Decision Letter · Decision Letter 2]

6 May 2026

Edge-Focused Wire Arc Additive Manufacturing: Method Development with ANN-Based Stress–Strain and Mass-Efficiency

PONE-D-25-54263R2

Dear Dr. Son Minh,

We’re pleased to inform you that your manuscript has been judged scientifically suitable for publication and will be formally accepted for publication once it meets all outstanding technical requirements.

Kind regards,

Antonio Riveiro Rodríguez, PhD

Academic Editor

PLOS One

Reviewers' comments:

Reviewer's Responses to Questions

**Comments to the Author**

1. If the authors have adequately addressed your comments raised in a previous round of review and you feel that this manuscript is now acceptable for publication, you may indicate that here to bypass the “Comments to the Author” section, enter your conflict of interest statement in the “Confidential to Editor” section, and submit your "Accept" recommendation.

Reviewer #2: All comments have been addressed

2. Is the manuscript technically sound, and do the data support the conclusions?

Reviewer #2: Yes

3. Has the statistical analysis been performed appropriately and rigorously? 

Reviewer #2: Yes

4. Have the authors made all data underlying the findings in their manuscript fully available?

Reviewer #2: Yes

5. Is the manuscript presented in an intelligible fashion and written in standard English?

Reviewer #2: Yes

6. Review Comments to the Author

Reviewer #2: All comments had been been successfully addressed by the authors. Congratulation to the authors for the effort made.

7. PLOS authors have the option to publish the peer review history of their article (what does this mean?). If published, this will include your full peer review and any attached files.

Reviewer #2: No

---

## [Editor Report · Acceptance letter]

PONE-D-25-54263R2

PLOS One

Dear Dr. Son Minh,

I'm pleased to inform you that your manuscript has been deemed suitable for publication in PLOS One. Congratulations! Your manuscript is now being handed over to our production team.

Kind regards,

on behalf of

Dr. Antonio Riveiro Rodríguez

Academic Editor

PLOS One